# Mechanistic insight for T-cell exclusion by cancer-associated fibroblasts in human lung cancer

Joseph Ackermann[1,2], Chiara Bernard[3], Philemon Sirven[4], Helene Salmon[4], Massimiliano Fraldi[2,3], Martine D Ben Amar[2,5]*

[1]Laboratoire Jean Perrin, Sorbonne Université, Paris, France; [2]Laboratoire de Physique de l'Ecole normale supérieure, ENS, Université PSL, CNRS, Sorbonne Université, Université Paris Cité, Paris, France; [3]Department of Structures for Engineering and Architecture, University of Naples "Federico II", Naples, Italy; [4]Institut Curie, PSL Research University, INSERM, Paris, France; [5]Institut Universitaire de Cancérologie, Faculté de médecine, Sorbonne Université, Paris, France

*For correspondence: benamar@lps.ens.fr

## eLife Assessment

This is a **valuable** report of a spatially-extended model to study the complex interactions between immune cells, fibroblasts, and cancer cells, providing insights into how fibroblast activation can influence tumor progression. The model opens up new possibilities for studying fibroblast-driven effects in diverse settings, which is crucial for understanding potential tumor microenvironment manipulations that could enhance immunotherapy efficacy. While the results presented are **convincing** and follow logically from the model's assumptions, some of these assumptions, as acknowledged by the authors, may oversimplify certain aspects in light of complex experimental findings, system geometry, and general principles of active matter research. Nonetheless, the authors provide justification for their work as a meaningful step towards more comprehensive modeling approaches.

**Abstract** The tumor stroma consists mainly of extracellular matrix, fibroblasts, immune cells, and vasculature. Its structure and functions are altered during malignancy: tumor cells transform fibroblasts into cancer-associated fibroblasts, which exhibit immunosuppressive activities on which growth and metastasis depend. These include exclusion of immune cells from the tumor nest, cancer progression, and inhibition of T-cell-based immunotherapy. To understand these complex interactions, we measure the density of different cell types in the stroma using immunohistochemistry techniques on tumor samples from lung cancer patients. We incorporate these data into a minimal dynamical system, explore the variety of outcomes, and finally establish a spatio-temporal model that explains the cell distribution. We reproduce that cancer-associated fibroblasts act as a barrier to tumor expansion, but also reduce the efficiency of the immune response. Our conclusion is that the final outcome depends on the parameter values for each patient and leads to either tumor invasion, persistence, or eradication as a result of the interplay between cancer cell growth, T-cell cytotoxicity, and fibroblast activity. However, despite the existence of a wide range of scenarios, distinct trajectories, and patterns allow quantitative predictions that may help in the selection of new therapies and personalized protocols.

## Introduction

Cancer results from the malignant transformation of cells due to genetic changes or damage that causes cells to grow and spread in an abnormal and uncontrolled way. Mutated cells can form solid tumor tissue at specific sites and spread to distant regions of the body through metastasis. The growth, development, and response to drugs and therapies of tumor masses are highly dependent on biophysical environmental conditions. These involve a not fully understood crosstalk between cancer cells and the surrounding stroma, which consists of cancer-associated fibroblasts, vessels, and immune cells embedded in the extracellular matrix. Indeed, the tumor-stroma ratio turns out to be an independent prognostic factor, with a large proportion of stroma leading to a worse prognosis (*Wu et al., 2016*). In this paper, we focus on modeling tumor-stroma interactions in lung cancer which is one of the most common and deadly cancers, with more than 2.2 million new cases diagnosed and 1.8 million deaths worldwide in 2020. Non-small cell lung carcinoma (NSCLC) accounts for the vast majority of lung cancers (85%) (*Sung et al., 2021*). The majority of cells that make up the stroma are fibroblasts and macrophages. Both types are affected and reprogrammed by cancer cells and have been shown to act as tumor promoters and tumor suppressors, depending on the stage of tumor progression and of its mutational status (*Schreiber et al., 2011*; *Coussens and Werb, 2002*). As cancer progresses, tumor cells often transform surrounding healthy fibroblasts into cancer-associated fibroblasts (CAFs) which, similarly to a wound healing context, produce higher levels of extracellular matrix (ECM) as well as growth factors and cytokines that affect the recruitment of immune or vascular cells and the growth of cancer cells (*Mathieson et al., 2022*).

Experimental observations and evidence of the mutual interactions between cancer cells, the immune response, and cancer-associated fibroblasts suggest that a multi-physical model of the tumor microenvironment (TME), including molecular and spatial dynamics, could be highly beneficial in understanding and predicting the complex evolution of tumor growth and progression. Thus, such analysis may represent a critical step in improving T-cell-based therapies. Quantitative models already exist for biochemical interactions between fibroblasts and cancer cells (*Kim et al., 2010*; *Kim and Friedman, 2010*; *Heidary et al., 2020*), or for the T-cell recruitment in immunotherapy (*Ruiz-Martinez et al., 2022*), but models for multi-species are less common. A recent paper by Mukherjee et al. analyzed the spatial dynamics of infiltrating splenocytes in an aggregate of cancerous melanocytes using experimental data obtained in vitro (*Mukherjee et al., 2023*). Their conclusion is that a strong persistent random walk and contact energies are important for the ability of T-cells to infiltrate the tumor. However, to the best of our knowledge, there are no approaches in the current literature to model the dynamics behind human lung tumor proliferation due to the role of CAFs in excluding T-cells from the nest. Here, we propose such a model for NSCLC tumors to describe the dynamic interplay between cancer cells, T-cells, and fibroblasts in their non-activated and activated states, incorporating the most relevant interactions within the human tumor microenvironment.

To this end, we briefly review the literature on the interactions between different tumor components, including quantitative information on their composition. In particular, we base our model on data extracted from immuno-histochemistry (IHC) staining of the tumor microenvironment performed on a large cohort of human NSCLC tumor samples. This allows us to calibrate our model so that the densities and localization of the various species that the model yields are consistent with the actual data (*Grout et al., 2022*) (Section Lung Cancer Microenvironment). We then build the theoretical description in two steps. In a first step, we ignore any spatial organization and model the proliferation and activation phenomena alone, while introducing a pressure-like term to avoid excessive proliferation. This leads to a nonlinear dynamical system for the cell concentrations (Section Dynamic Modeling in the Lung Cancer TME). Careful analysis of this dynamical system yields a stable steady state in which T-cells become inefficient against tumor survival for a given set of parameters. More precisely, we demonstrate that two mechanisms control tumor growth: the competition with other species for resources and space, and the efficiency of the T-cell response. In a second step, we enrich the model by considering the spatio-temporal evolution of tumor growth through a continuum approach for cell mixtures based on Onsager's variational principle. Indeed, it has been shown that the spatial distribution of T-cells inside the tumor is an indicator of the patient survival (*Carstens et al., 2017*). Finite element-based simulations in two-dimensional space demonstrate the ambiguous activity of CAFs in regulating cancer cell proliferation and invasion. Finally, we discuss our results obtained with this minimal model and suggest directions for future works, such as considering more precisely the

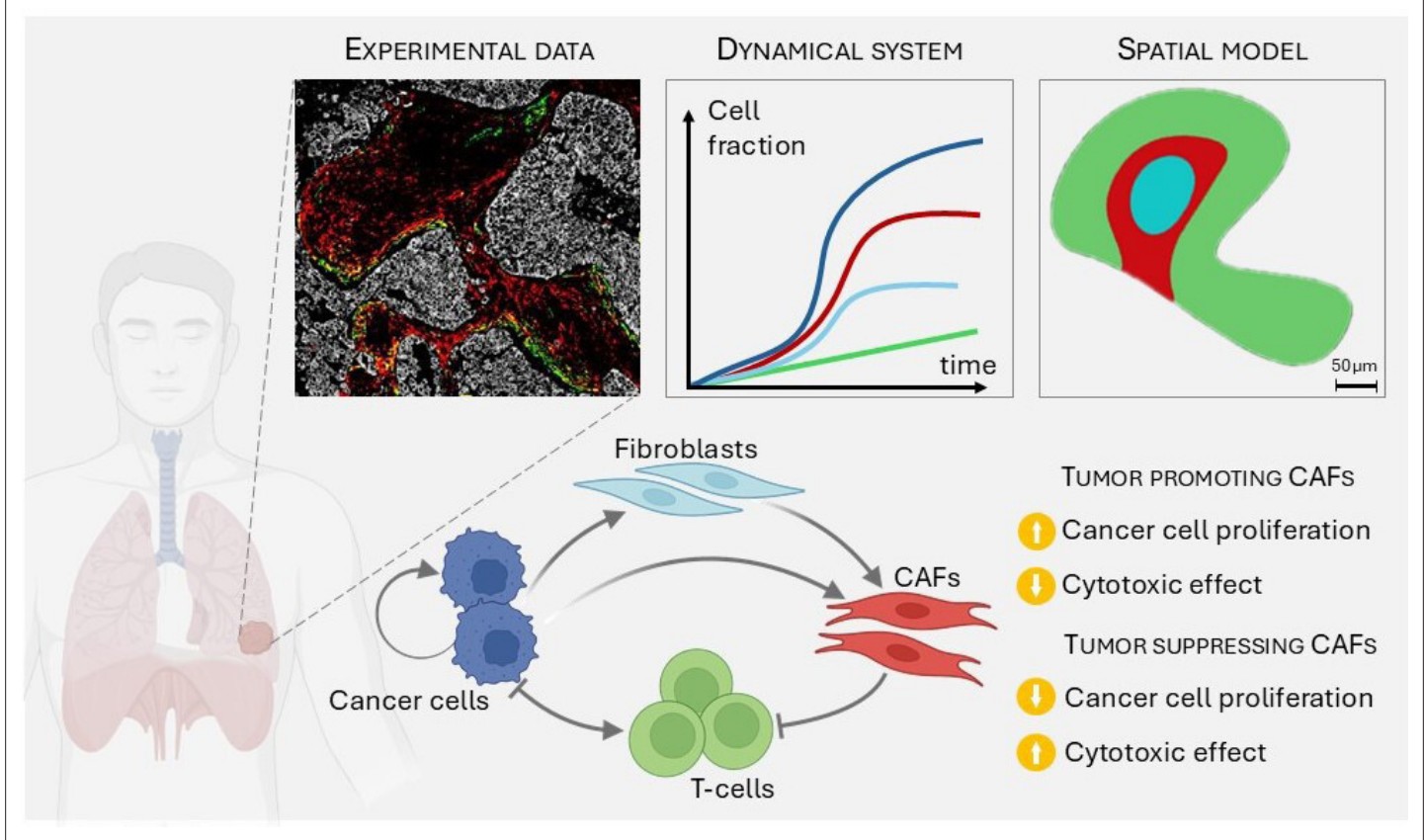

**Figure 1.** Summary of the article: Starting from experimental data on the structure and density of the different species on lung cancer, we first build a dynamical system, and then a spatiotemporal model. We incorporate four species with specific interactions, that may lead to different outcomes.

geometry of the tumor micro-environment, and introducing anisotropic friction that would depend on the nematic orientation of the fibers (Section Spatio-Temporal Behavior of Tumor Growth). In the Appendix, further details about the results of numerical simulations are provided to show how the proposed dynamical system is able to capture different evolutionary and tumor fates depending on the role played by the species, their initial conditions, as well as the shapes and distributions of the regions occupied by cells. In the Supplementary Information (SI), section A provides more details for the calculation of the dynamical system section, and section B for the anisotropic friction. A summary of the article is provided in *Figure 1*.

## Lung cancer microenvironment

Modeling the tumor environment requires a deep understanding of the interactions between the various components, such as cells of different types, soft materials, and fibers as well as fluids and diffusing molecules. These interactions are summarized in section Interactions Between TME Components. Next, in section Lung TME Composition, we present data on the density of the different cell types in the TME. Indeed, our continuous approach incorporates in vivo biological data that vary between patients and tumor sites, and also evolve over time within the tumor. Therefore, after a review of the available values found in the literature, we performed direct measurements from samples stained by multiplex IHC to study the composition of fibroblasts and distribution of T-cells in human NSCLC tumors in situ (*Grout et al., 2022*).

### Interactions between TME components

Almost all tissues contain a population of fibroblasts that provide the tissue architecture, and serve as sentinels for tissue dysfunctions. When a solid tumor grows, quiescent or progenitor lung fibroblasts activate an initial wound-healing response with increased production of ECM and growth factor

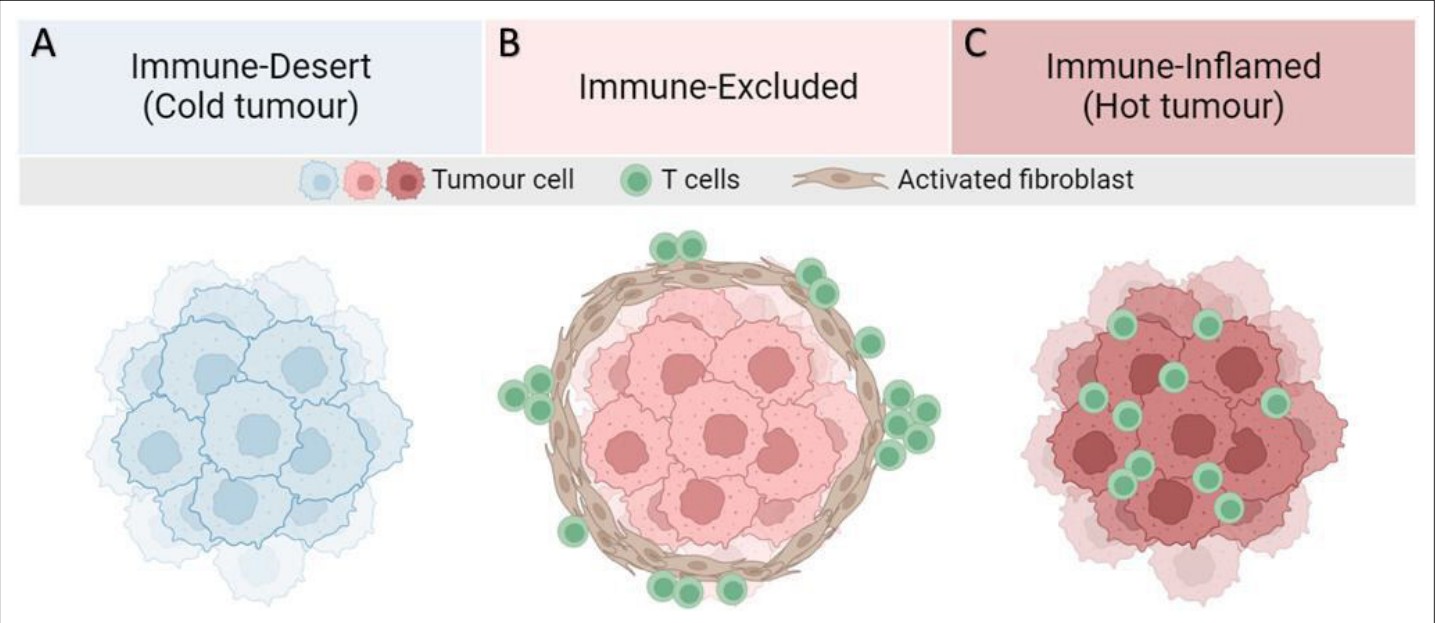

**Figure 2.** Major T-cell infiltration patterns observed in solid tumors. (**A**) Lack of tumor antigen, inadequate priming, defects in antigen presentation and/or lack of presentation, and/or lack of T-cell-attracting chemokines result in the absence of T-cells in the tumor. (**B**) Presence of T-cells in invasive margins but absent in the tumor bed. Immune evasion may be due to stromal barriers, lack of chemokines, aberrant vasculature, or hypoxia. (**C**) High degree of T-cell infiltration forms a hot tumor.

(cytokine) and upregulation of activation markers such as FAP (**Salmon et al., 2019**). Due to mechano-transduction and/or biochemical signals from tumor or immune cells, CAFs often increase their level of contractility, which affects the maintenance of the tumor stroma (**Dvorak, 1986**). CAFs are characterized by their increased mobility, proliferation, and ECM remodeling and, unlike wound-associated fibroblasts, they seem to undergo poorly reversible activation in absence of an appropriate therapy (**Nakamura et al., 2004**; **Wang et al., 2009**; **Peng et al., 2013**; **Kalluri, 2016**). The diversity of this population, resulting from phenotypic modifications, explains their diverse functions and localization in the stroma, close or distant from the tumor nest (**Barrett and Puré, 2020**; **Grout et al., 2022**).

Activated fibroblasts produce fibers that act as a mechanical barrier around the tumor, impeding the movement of immune cells and limiting the interaction between cytotoxic T-cells and cancer cells (**Salmon et al., 2012**; **Yaegashi et al., 2021**). However, by creating such a barrier around tumors, CAFs may also prevent the spread of cancer, as mechanical stress can reduce cell spreading and promote cell apoptosis (**Cheng et al., 2009**; **Barbazan et al., 2023**). In addition, biochemical factors expressed by CAFs also help to modify the phenotype of T-cells or inactivate their cytotoxic capacity (**Nazareth et al., 2007**; **Lakins et al., 2018**; **Koppensteiner et al., 2022**). Thus, CAFs have an ambiguous role as tumor promoters, inhibiting T-cell invasion into the nest, and as tumor suppressors, limiting the cancer growth and giving rise to an immune-excluded tumor (**Figure 2B**), which is less proliferative than a free-growing tumor (**Figure 2A**). Failure to confine the tumor induces high levels of cytotoxic T-cell infiltration, creating a hot tumor (**Figure 2C**). Therefore, one aim of this study is to predict susceptibility of a tumor to T-cell infiltration according to the scheme given in **Figure 2**.

In the context of non-small cell lung carcinoma (NSCLC), the recruitment of CD8 [+] T-cells seems to be modulated by a specific tumor-associated antigen present on the surface of the cancer cells (**Hiraoka et al., 2006**). In particular, in the family of inflammatory proteins, also called cytokines, Interlukin-6 (IL6) and (IL8) seem, among others to stimulate the infiltration of CD8[+] (**Liu et al., 2022**; **McKeown et al., 2004**; **Brenner et al., 2017**). The origins of CD8[+] T-cells that infiltrate the tumor are diverse, as they differentiate from both circulating and tissue-resident precursors (**Gueguen et al., 2021**). However, infiltration alone does not imply that the immune response is efficient. In fact, an immunotherapy such as anti-PD1/PDL1 antibodies is often needed to boost the response of these T-cells. Chemical factors also play an important role in attracting T-cells via chemotaxis, such as the

chemokines produced by dendritic cells or cancer cells (*Brown et al., 2007*). The absence of such chemicals leads to the formation of the so-called immune-desert tumor (*Figure 2A*).

## Lung TME composition

For a quantitative modeling, it is critical to use biologically well-identified data on both tumor and stroma, in terms of the density and the localization of the different species. Therefore, we survey the literature to build a quantitative view for the composition of the TME. We recapitulate these values in *Table 1*. However, human data show a great diversity depending on many factors such as tumor edge, patient age, non-cancer health status, as well as method of analysis. The fibroblast population in the stroma, which is very heterogeneous and not easy to identify, is the best example. In particular, although fibroblasts are often considered to be a major component of the stroma, they are only found in small proportions in scRNA-seq with 10 X single-cell systems, which may be rather surprising. This situation is well known and may be due to tissue digestion processes, to the fact that part of the stroma is not extracted with the tumor nodule, and to a lower efficiency of 10 X single-cell systems for fibroblasts. Another caveat is that identifying the surface fraction of cells by staining can be very different from counting individual cells. In fact, cell types can vary greatly in size. For example, lung cancer cells have a diameter between 13–18 µm (*Hosokawa et al., 2013*), fibroblasts ~16 µm (*Mitsui and Schneider, 1976*), and T-cells between 5–10 µm (*Tasnim et al., 2018*), resulting in volumes and projected areas that can be 10 times smaller for T-cells compared to fibroblasts and cancer cells for the same densities. Moreover, in some studies, authors focus on well-characterized zones of enrichment for the different species (tumor nest, fibrotic areas), and do not consider each cell type per se (*Sieren et al., 2010*; *Sieren et al., 2011*). Despite these remarks, cancer cells, T-cells, and α TME. The limitations of using data coming from the literature led us to perform our own measurements. In this study, we analyzed data from 13 patients with lung squamous cell carcinoma (LUSC) and 22 patients with lung adenocarcinoma (LUAD), based on a recent publication by some of the present authors (*Grout*

**Table 1.** Composition of the tumor microenvironment.

This table presents the literature about the fraction of different species in the lung TME. The values found in the present studies are written in the two last rows (raw data is provided as Supplementary Material). Different methods have been used, in different subtypes of lung cancer. We introduce the following abbreviations. (C-c): Cancer cells. (MΦ): Macrophages. (T-c): T-cells. (Fb): Fibroblasts. (scRNA-seq): single-cell RNA seq,(NSCLC): Non Small Cell Lung Cancer, (LUAD): Adenocarcinoma, (SSN): Sub-Solid Nodule. (S): stroma region. (T): tumor nest region. (ST): Stroma + tumor nest region. (TSR): tumor stroma ratio. Data for Ref (*Ireland et al., 2020*; *Kim et al., 2020*; *Laughney et al., 2020*; *Maynard et al., 2020*; *Qian et al., 2020*) were extracted from *Curated Cancer Cell Atlas, 2023*.

| Method | Sample | C-c (%) | T-c (%) | Fb (%) | MΦ(%) | TSR |
|---|---|---|---|---|---|---|
| Staining *Sieren et al., 2010* | LUAD | >25 | | >20 | | 0.3 |
| Staining *Sieren et al., 2011* | LUAD | 67 | | >6.5 | | 2.6 |
| scRNA-seq *Lambrechts et al., 2018* | NSCLC | | 55 (S) | 4 (S) | 15 (S) | |
| scRNA-seq *Ireland et al., 2020* | SCLC | 76 | | | | |
| scRNA-seq *Kim et al., 2020* | LUAD | 33 | 19.5 | ~6 | 20.4 | |
| scRNA-seq *Laughney et al., 2020* | LUAD | <12 | 42.6 | ~1 | 12.2 | |
| scRNA-seq *Maynard et al., 2020* | NSCLC | <22.7 | 16 | 8 | 23 | |
| scRNA-seq *Qian et al., 2020* | mixed | 19.9 | 28.1 | ~2 | 27.5 | |
| scRNA-seq *Xing et al., 2021* | LUAD | 16.4 | 30 | ~2 | 18.4 | |
| scRNA-seq *Altorki et al., 2022* | LUAD | 12 | 7.5 | | | |
| Staining *Mathieson et al., 2022* | NSCLC | | | 75 (S), 0 (T) | | |
| Staining (present article) | LUAD | 47±13 | | 31±17 (S), 0 (T) | | 1.07±0.68 |
| Staining (present article) | LUSC | 54±10 | | 35±12 (S), 0(T) | | 1.30±0.53 |

The online version of this article includes the following source data for table 1:

**Source data 1.** Raw data for the two last lines of the table.

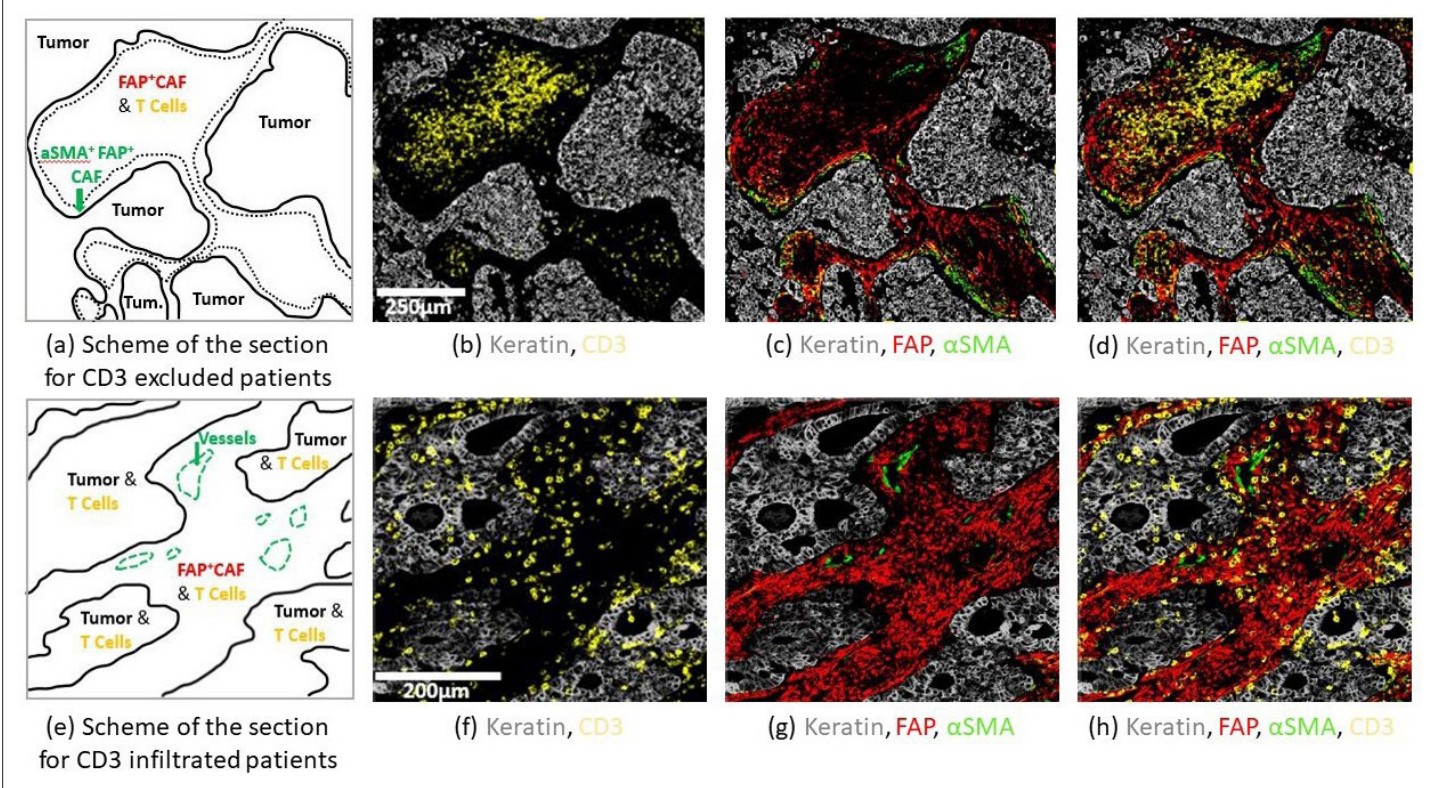

**Figure 3.** Structural organization in human Non Small Cell Lung Cancer (NSCLC). Staining was performed by multiplex immuno-histochemistry (IHC). FFPE NSCLC sections were stained for keratin as a marker of cancer cells (gray), CD3 (yellow), and fibroblast markers αSMA (green), FAP (red). First row: CD3 excluded patient. (**a**) Scheme of the section showing CD3+ cell exclusion from tumor nests. The green arrow highlights border regions with contractile fibroblast barrier αSMA+ FAP+ and low CD3+ cells. (**b**) CD3+ cells are localized in the center of the stroma. (**c**) Dense αSMA staining at the tumor border are associated with a decrease of CD3+ cell abundance. (**d**) Recap of all the markers. Second row: CD3 infiltrated patient. (**e**) Scheme of the section showing CD3+ infiltration in the tumor islets. The green arrow shows αSMA+ staining on vessels. (**f**) CD3+ are localized in the tumor nest. (**g**) FAP+ staining is localized throughout the stroma. (**h**) Recap of all the markers.

*et al., 2022*). In this previous study, different fibroblast types were identified using multiplex IHC imaging. Tumor islets were stained with keratin, T-cells with CD3, and fibroblasts with αSMA, FAP, and ADH1B. The coverage was evaluated for each fibroblast type and the total fibroblast coverage corresponds to the sum of these different coverages. Here, in the case of LUSC, we found that fibroblasts (composed of 6% ADH1B, 37% FAP+ αSMA, 25% FAP+ αSMA+, 31% FAP+ αSMA+) occupy 35% of the stroma with a tumor stroma ratio (TSR) of 1.29. In LUAD, these proportions were partly modified: fibroblasts (composed of 48% ADH1B, 18% FAP+ αSMA−, 7% FAP++ αSMA+, 26% FAP− αSMA+) occupy 31% of the stroma, with a TSR of 1.07. The fibroblasts most responsible for T-cell exclusion are shown to be FAP+ αSMA+(*Grout et al., 2022*). In *Figure 3*, we compare two microenvironments. In the top one, FAP+ αSMA+ are lining the tumor (*Figure 3a, c, d*), and T-cells are excluded from it (*Figure 3b*). On the contrary, in cases where no FAP+ αSMA+ are observed, fibroblasts are homogeneously distributed in the stroma (*Figure 3g, e, h*), and T-cells infiltrate the tumor (*Figure 3f*). Although no data were directly obtained for the surface fraction of T-cells, the estimated number of CD3+ T-cells measured in both tumor islands and stroma suggests a low surface fraction compared to cancer cells and fibroblasts (see Appendix C, Raw data for *Table 1*).

## Dynamic modeling in the lung cancer TME

Our theoretical and numerical analyses consist of two steps in the spirit of the article of *Olmeda and Ben Amar, 2019* for the study of cancer stem cells. In a first analysis, we examine only the dynamics of an interactive ecological system in order to evaluate the physical parameters that quantify these interactions and how the dynamics depend on them. Spatial constraints are represented by a pressure term avoiding overcrowding. We present this approach step by step in order to set the parameters

one by one, highlighting the physical importance of each choice through stability analyzes of the system. In addition, this first step allows to explore efficiently the parameter space without time-consuming numerical resolution, and to already produce a first classification of the global outcomes. The second step is the spatial description of the tumor growth in Section Spatio-Temporal Behavior of Tumor Growth. In both approaches, the case where fibroblasts, cancer cells, and T-cells are present is intended to correspond to the patient data presented in the previous section.

## Dynamical system for immune and cancer cells in interaction

As we saw in the previous section, the complexity of the microenvironment makes the role of the immune system hardly predictable and highly dependent on the tumor being studied. In the case of lung tumors, the immune system is triggered as the carcinoma expands, but T-cells may be excluded from the tumor nest by activated fibroblasts. Therefore, the goal of this work is to physically and quantitatively understand the process of T-cell exclusion from the tumor mass in the simplest way possible, and to explore different possible scenarios, since the dynamical system can be seen as the spatial integration of the different processes over the domain under study.

We focus on the interaction between different cell types in the case of the NSCLC. In particular, we consider a closed system including the cancer cells, T-cells, non-activated fibroblasts (NAFs), cancer-associated fibroblasts (CAFs) and healthy cells with the extracellular medium. Diffusive signaling molecules are not explicitly introduced: their production by one cell type and their effect on another cell type is modeled as a direct interaction between the two. For example, the attraction of T-cells to cancer cells by chemotaxis is introduced in the mathematical model as a source term proportional to the product of the two concentrations $T$ and $C$ in the T-cell equation (see below *Equation 2*). We also hypothesize that the main difference between NAFs and CAFs is the fiber production of the latter, which prevents T-cells from infiltrating the tumor. Furthermore, our model does not consider transformation of NAFs into CAFs as a reversible process. All these cells have the same mass density and the sum of their mass fraction satisfies the relationship $\mathcal{S} = C + T + F_{NA} + F_A = 1 - N$, where $N$ is a healthy non active component such as healthy cells and interstitial fluid, for example. The mass fraction of cancer cells is represented by $C$, T-cells by $T$, quiescent or non-activated fibroblasts NAFs by $F_{NA}$, activated fibroblasts CAFs by $F_A$. Note that $N$ which is not an active component does not appear in the following equations. Also, we do not consider here the recruitment of macrophages to better highlight the competing mechanisms related to the sole role of T-cells and CAFs in the tumor mass development. With this in mind, we write an evolution equation for each component of the system: $dX/dt = \Gamma_X$, where $\Gamma_X$ corresponds to a source term modeling the proliferation, death, differentiation, or fluxes into/out of the system under study. The source terms for each species are described in detail below.

The dynamics of the cancer cells are driven by their proliferation, controlled by a growth rate coefficient $\alpha_C$, and limited by a death rate $\tilde{\delta}_C$. It takes into account the population pressure caused by self-inhibition as well as by the inhibition of the other species. In the following, we will choose $\alpha_C^{-1}$ as the unit of time, and all coefficients introduced in the following will be pure constants without unit, so that $\tilde{\delta}_C = \alpha_C \delta_C$. In addition, cytotoxic T-cells eliminate cancer cells if their anti-tumor activity is not inhibited by the activated fibroblasts (although they do not remove T-cells from the mixture). This process is quantified by the cytotoxic coefficient $\delta_{CT}$, and by the coefficient of T-cell inhibition by CAFs, $\delta_{TF}$. With these assumptions, the dynamics of the cancer cell mass fraction can be read:

$$\frac{dC}{dt} = C - \frac{\delta_{CT} CT}{1 + \delta_{TF} F_A} - \delta_C C \mathcal{S}. \tag{1}$$

Although proliferation of cytotoxic T-cells has been observed, we do not consider explicitly proliferation in our study as we focus on their ability to infiltrate the tumor. Rather, we consider that T-cells proliferate outside the domain boundaries, so that this proliferation is included in the boundary source contributions. Therefore, their only source is their attraction to cancer cells, which occurs at a recruitment rate of $\alpha_{TC} C$, while the crowding limits this recruitment:

$$\frac{dT}{dt} = \alpha_{TC} C - \delta_T T \mathcal{S}. \tag{2}$$

Fibroblasts are the key regulators of tumor immunity and progression. Their dynamics involve a recruitment of non-activated fibroblasts $\alpha_{NA}$ whose role is to maintain an adequate supply given by $\alpha_{NA}/\delta_{NA}$ in healthy tissue, i.e., in the absence of cancer cells, and a death rate due to the pressure exerted by the cells and controlled by $\delta_{NA}$. NAFs are attracted to the tumor nest at a rate $\alpha_{NA,C}C$ by the cancer cells that activate them to become CAFs. This process is accounted by introducing a specific transformation rate controlled by the plasticity coefficient $K_A$, so that the dynamic equation for NAFs reads:

$$\frac{dF_{NA}}{dt} = \alpha_{NA} + \alpha_{NA,C}C - K_A F_{NA}C - \delta_{NA}F_{NA}\mathcal{S}. \tag{3}$$

Thus, the dynamics of the CAFs is purely determined by the transformation of the NAFs and by the pressure through $\delta_A$, which leads to:

$$\frac{dF_A}{dt} = K_A F_{NA}C - \delta_A F_A \mathcal{S}. \tag{4}$$

Therefore, the role of fibroblasts in human lung carcinoma can be investigated by studying the interplay between cancer cells and the TME. The interactions discussed above are described by 10 parameters and lead to a system of four coupled nonlinear differential equations concerning four unknowns. Within this framework, we proceed to the estimation of the parameters and the steady states of the system.

## Model parameters and fixed point analysis

When all the parameters vary, the steady states or the equilibrium points of the dynamics can be quite impractical, so we start by assuming that all the coefficients at the origin of a pressure are equivalent: $\delta = \delta_C = \delta_T = \delta_A = \delta_{NA} > 0$. This reduces the number of independent parameters to 7. Fixed points are obtained by setting the time derivatives in *Equations 1–4* to 0 which gives the long-term behavior when the system reaches equilibrium.

In order to evaluate the values of the parameters in *Equations 1–4*, we study different simplified situations that can be reproduced in experiments in vitro or in controlled experiments in vivo. We also use the conclusions obtained in Section Lung TME Composition for the cell densities. We will start by analyzing the case of cancer cells alone and we will successively add all the other cell types with fractions $C, T, F_{NA}, F_A$. More details on the calculations can be found in Appendix 1, Section A, *Dynamical system*.

### Cancer cells and T-cells

The dynamic evolution of cancer cells alone is limited to: $dC/dt = C - \delta C^2$. This equation leads to two equilibrium fixed points: $C = 0$ and $C = \delta^{-1}$ and only the second one $C = \delta^{-1}$ is stable. When cancer cells are isolated from other active cellular components, they are expected to invade the system or to be its major component, leading to a cancer death rate of $\delta \sim 1$.

We now examine the interaction between T-cells and cancer cells in the nest in the absence of fibroblasts (so $F_{NA} = F_A = 0$) and the relevant parameters scaled by $\delta$ are $\delta_{CT}$ and $\alpha_{TC}$. Then, we study the equilibrium regime:

$$\begin{cases} \dfrac{dC}{dt} = 0 = C - \delta_{CT}CT - \delta C(C+T), \\ \dfrac{dT}{dt} = 0 = \alpha_{TC}C - \delta T(C+T). \end{cases} \tag{5}$$

There are three equilibrium solution pairs $C, T$, including the trivial one: $\{0, 0\}$. To analyze whether the solutions found are physically relevant ($0 \le C, T \le 1$) and dynamically stable, we estimate possible scaling for the two parameters, focusing on an effective immune response against cancer. For efficient elimination of cancer cells by T-cells, the killing rate $\delta_{CT}$ must be much larger than the natural death rate $\delta$, so we introduce a small parameter $0 < \epsilon \ll 1$, so that $\delta_{CT} = \delta\epsilon^{-1}$. We also assume that the T-cell recruitment is slow compared to cancer cells, which means that $\alpha_{TC} = a_0\epsilon$, where $a_0$ being of order one. So the only stable solution is $\{C_+, T_+\} = \dfrac{\epsilon}{\delta}\{\dfrac{1}{a_0-1}, 1\}$ where $a_0 > 1$ (see SI, Interaction between

T-cells and cancer cells). Thus, even if the inflammation level is low, resulting in a small number of T-cells, the immune action on the cancer cells remains efficient.

## Role of activated fibroblasts on T-cells

Fibroblasts play an active role in the exclusion of the T-cells from the tumor nest. We isolate a subsystem composed of cancer cells, T-cells, and active fibroblasts to determine the inhibition rate $\delta_{TF}$, responsible for the marginalization of the T-cells and subsequently for the increase of cancer cells at a fixed volume fraction of fibroblasts. We consider the fixed points corresponding to an ensemble $\{C, T, 0, F_A\}$. For simplicity, we first assume that $F_A$ is constant, which leads to the dynamical system:

$$\begin{cases} \dfrac{dC}{dt} = 0 = f_0 C - \Delta_{CT} CT - \delta C(C+T), \\ \dfrac{dT}{dt} = 0 = \alpha_{TC} C - \Delta_F T - \delta T(C+T), \end{cases}$$

where $f_0 = 1 - \delta F_A$, $\Delta_{CT} = \delta_{CT}(1 + \delta_{TF}F_A)^{-1}$ and $\Delta_F = \delta F_A$. There are four solution pairs (see SI, Role of activated fibroblasts on T-cells).

We use the scaling of both parameters already established in the previous paragraph: $\delta_{CT} = \delta\epsilon^{-1}, \alpha_{TC} = a_0\epsilon$. We also assume that the inhibition of the T-cells may counteract their cytotoxic effect on cancer cells, i.e., $(1 + \delta_{TF}F_A)^{-1} \sim \epsilon$, when activated fibroblasts are abundant $F_A = \delta^{-1}f_a$. This results in $\delta_{TF} = d_0\delta\epsilon^{-1}$.

The only stable equilibrium solution is $(C_+, T_+) = (f_0\delta^{-1} + \mathcal{O}(\epsilon), \mathcal{O}(\epsilon))$, where the notation $\mathcal{O}(\epsilon)$ denotes a quantity whose order of magnitude is $\epsilon$. Thus, when fibroblasts inactivate T-cells, they promote the cancer cell proliferation.

## Residual fibroblasts in healthy tissue

In healthy tissue, most of the fibroblasts are in a quiescent state and are not activated in the absence of pathologies such as wounds, allergic reactions, or cancer cells. Therefore, in such tissues, for a quiescent fibroblast population, the density of $F_{NA}$ is the equilibrium solution of the equation $dF_{NA}/dt = 0 = \alpha_{NA} - \delta F_{NA}^2$. The parameter $\alpha_{NA}$ represents the net influx of fibroblasts into the tissue. Assuming the fraction of non-activated fibroblasts to be residual $\sim \epsilon^2$, we deduce $\alpha_{NA} \sim \delta\epsilon^4$.

## Fibroblast plasticity

Fibroblast plasticity is the phenotypic change responsible for T-cell inhibition and for more active fiber production. Cancer cells drive this transformation of the current population $F_{NA}$ into $F_A$. This process is quantified by the constant $K_A$. We first estimate that the fibroblast population is comparable to the cancer cell population when they are alone, which leads to: $F_{NA} = f_n\delta^{-1}$ where $f_n \sim 1$ is a constant.

Replacing $\delta_{CT} = \delta\epsilon^{-1}, \alpha_{TC} = a_0\epsilon, \delta_{TF} = d_0\delta\epsilon^{-1}$, leads to:

$$\begin{cases} \dfrac{dC}{dt} = 0 = C - \dfrac{\delta\epsilon^{-1}CT}{1 + d_0\delta\epsilon^{-1}F_A} - \delta C\mathcal{S}, \\ \dfrac{dT}{dt} = 0 = a_0\epsilon C - \delta T\mathcal{S}, \\ \dfrac{dF_A}{dt} = 0 = K_A f_n C/\delta - \delta F_A\mathcal{S}. \end{cases} \tag{6}$$

Because of the effect of the fibroblasts on them, the T-cells have little effect on the cancer cells, so cancer cell proliferation is weakly affected. At equilibrium, this leads to $C \sim \delta^{-1}$, $T \sim \epsilon\delta^{-1}$. The equation for $T$ gives $F_A \sim a_0\delta^{-1}/d_0$. This means that the activated fibroblasts must be relatively abundant to fully inhibit the activity of the T-cells. It follows that the plasticity parameter is of low order in $\epsilon$, since the non-activated fibroblasts are maintained at a high density: $K_A \sim \delta$ (see SI, Fibroblast plasticity).

## Tumor fibroblast attraction

In this last step, we want to determine the parameter that controls the attraction of the fibroblasts to the tumor. To do this, we write $K_A = k\delta$ in the system of equations *Equations 1–4*, which we rewrite according to the previous findings:

$$\begin{cases} \dfrac{dC}{dt} = 0 = C - \dfrac{\delta\epsilon^{-1}CT}{1 + d_0\delta\epsilon^{-1}F_A} - \delta C\mathcal{S}, \\[2mm] \dfrac{dT}{dt} = 0 = a_0\epsilon C - \delta T\mathcal{S}, \\[2mm] \dfrac{dF_{NA}}{dt} = 0 = a_1\delta\epsilon^4 + \alpha_{NA,C}C - k\delta F_{NA}C - \delta F_{NA}\,\mathcal{S}, \\[2mm] \dfrac{dF_A}{dt} = 0 = k\delta F_{NA}C - \delta F_A\mathcal{S}. \end{cases} \tag{7}$$

Then, we look for solutions of the type: $\{C, T, F_{NA}, F_A\} = \delta^{-1}\{c_0, T_0\epsilon, f_{NA}, f_A\}$. The order of magnitude of the attraction parameter is then $\alpha_{NA,C} \sim 1$ (see SI, Non active fibroblasts attraction to tumor).

In the next section, we summarize all scaling laws possible, according to the obtained results and present the different scenarios related to the state of the tumor, shown in **Figure 2**.

## Numerical study of cell population dynamics

The dynamics of each cellular component of the mixture can be systematically studied according to the full set of parameters summarized in **Table 2**, with the corresponding orders of magnitude. Some of them can be considered as fixed in the system, i.e., they do not vary significantly between the different tumors. This category includes $\delta^{-1}$, the inverse of the free tumor cell density, and $\alpha_{NA}$, the NAF attraction parameter to healthy tissue. The other parameters can be studied as control parameters.

The simulated time-dependent densities of each cell type are displayed in **Figure 4** for different sets of parameters in the system of equations in **Equations 1–4**. At time $t = 0$, we assume small mass fractions of cancer cells, T-cells, activated and quiescent fibroblasts. Over time, the particular choice of a quadratic model allows the dynamics to reach a plateau for each cell type, confirming the stability of the fixed points found in Section Model Parameters and Fixed Point Analysis.

In the case of an immune-desert tumor, i.e., when T-cells are not attracted to the tumor nest or are unable to penetrate it (**Figure 4a–d**), cancer cell growth is not limited by the immune response. However, this growth saturates when it reaches a steady state due to the competition for space and resources between the different species and controlled by $\delta$ and $\alpha_{NA,C} + \alpha_{TC}$.

T-cells efficiency can be quantified through the parameter $\alpha_{TC}\delta_{CT}/(\alpha_{NA,C}K\delta_{TF})$ (**Figure 4i**). If the T-cells are efficient, the tumor is said to be immune-inflamed and several scenarios can be discussed. Their response is triggered by the proliferation of cancer cells. Thus, in the absence of CAF inhibition,

**Table 2.** Scaling variation and estimation of the coefficients according to different scenarios.

The scenarios are described in Section Model Parameters and Fixed Point Analysis and shown in **Figure 4**. The coefficients above are those introduced in **Equations 1–4**. The left column summarizes the different roles that T-cells can play in a cell mixture and the values of the coefficients of the mixture are listed in the following horizontal lines of the table. The scaling of $\delta$ is always: 1 , and $\alpha_{NA} \sim \delta\epsilon^4$.

| | $\delta_{CT}$ | $\alpha_{TC}$ | $\delta_{TF}$ | $\alpha_{NA,C}$ | $K_A$ |
|---|---|---|---|---|---|
| Control of: | Killing C-c by T-c | Attraction of T-c by C-c | Inhibition of T-c by F-c | Attraction of F-c to C-c | Activation of F-c |
| Efficient T-cells (T-c) but no attraction by (C-c) | $\delta\epsilon^{-1}$ $\sim 10$ | $\epsilon^2$ $\sim 0.01$ | No role | No role | No role |
| Efficient T-cells but inhibited by (F-c) | $\delta\epsilon^{-1}$ $\sim 10$ | $\epsilon$ $\sim 0.1$ | $\delta\epsilon^{-1}$ $\sim 10$ | $1{\sim}1$ | $\delta$ $\sim 1$ |
| Inefficient T-cellsno need of fibroblasts | $\delta$ $\leq 1$ | $\epsilon$ $\sim 0.1$ | No role | No role | No role |
| Efficient T-cells not inhibited by fibroblasts | $\delta\epsilon^{-1}$ $\sim 10$ | $\epsilon$ $\sim 0.1$ | $\delta$ $\sim 1$ | $1{\sim}1$ | $\delta$ $\sim 1$ |
| Efficient T-cells and fibroblasts not attracted | $\delta\epsilon^{-1}$ $\sim 10$ | $\epsilon$ $\sim 0.1$ | $\delta\epsilon^{-1}$ $\sim 10$ | $\epsilon$ $\sim 0.1$ | $\delta$ $\sim 1$ |
| Efficient T-cells fibroblasts not activated | $\delta\epsilon^{-1}$ $\sim 10$ | $\epsilon$ $\sim 0.1$ | $\delta\epsilon^{-1}$ $\sim 10$ | $\epsilon$ $\sim 1$ | $\delta\epsilon$ $\sim 0.1$ |

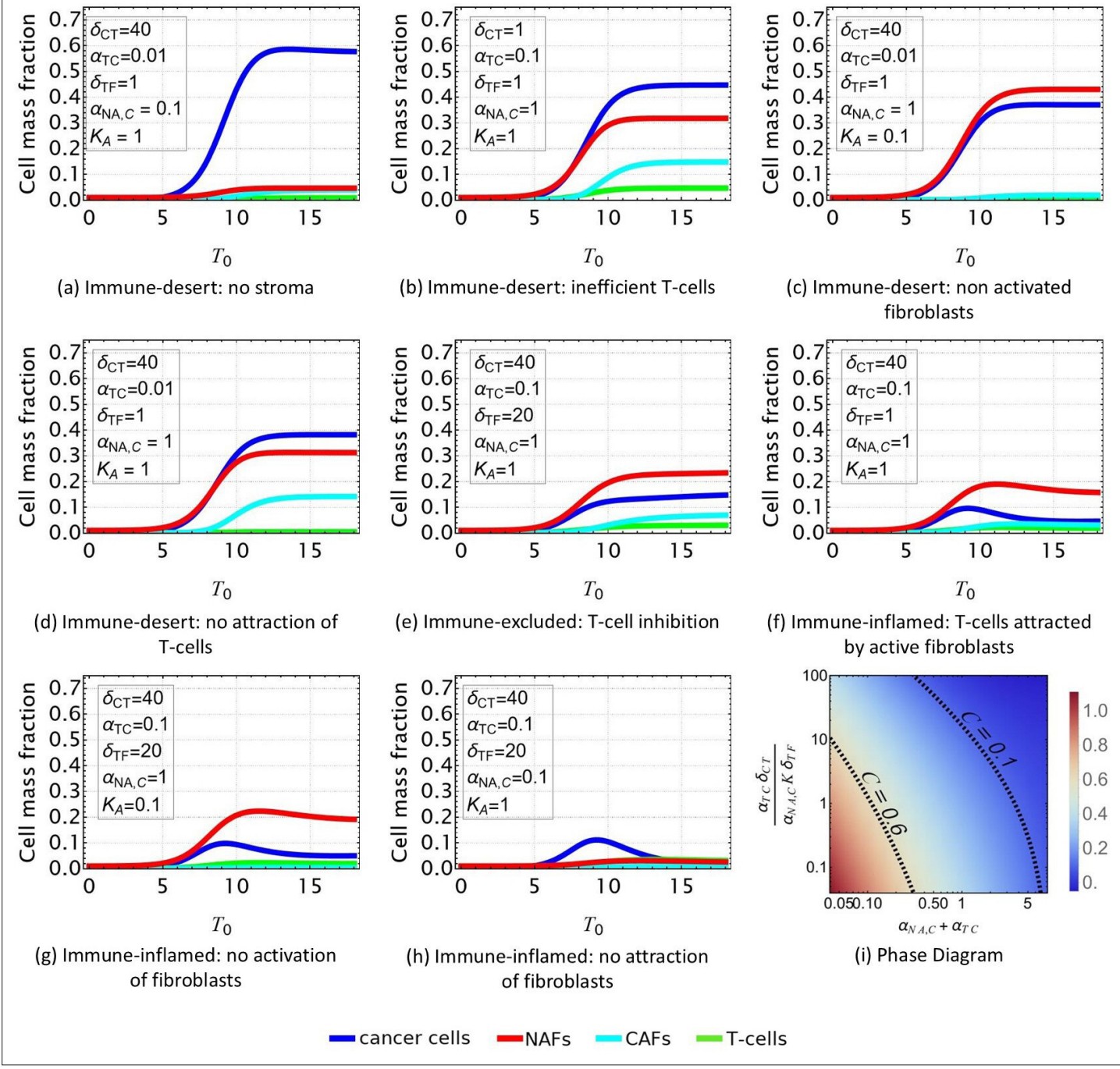

**Figure 4.** Evolution of the tumor microenvironment (TME) composition over time. Different profiles are obtained from *Equations 1–4* according to the set of parameters reported in *Table 2* and consistent with the scenarios shown in *Figure 2*. In all the plots, $\delta = 1$ and $\alpha_{NA} = 10^{-4}$. (a, b, c, d) Immune-desert tumor. Cancer cells proliferate when the immune response is inefficient or when T-cells are not attracted to the tumor. e: Immune-excluded tumor. The cancer-associated fibroblast (CAF) barrier inhibits T-cell infiltration, and promotes tumor growth. (f, g, h) Immune-inflamed tumor. T-cell infiltration limits cancer cell growth. (i) Density plot of the equilibrium cancer cell fraction at fixed $\delta$ and, $\alpha_{NA}$ in function of the two parameters that control the growth: $\alpha_{TC}\delta_{CT}/(\alpha_{NA,C}K\delta_{TF})$ reflects the ability of T-cells to kill the tumor, and $\alpha_{NA,C} + \alpha_{TC}$ reflects the competition between species for space and resources.

T-cells inhibit cancer growth (*Figure 4f–h*). In contrast, in the immune-excluded tumor, CAFs impede the T-cell response and thus indirectly promote cancer cell growth, as observed in *Figure 4e*.

In conclusion, our dynamical system, limited to four cell types, recapitulates the different possible scenarios that could evolve according to the interaction between cancer cell, T-cells, and fibroblasts in

lung adenocarcinoma. With respect to cancer cells, three main outcomes can be identified: the tumor can invade the tissue (*Figure 4a*), the cancer cell population can be limited without disappearing (*Figure 4b–e*), or it can be eradicated (*Figure 4f–h*). Each outcome can have multiple origins, that are all related to either competition for space and resources or T-cell efficiency, see *Figure 4i*. Each origin has a particular signature with respect to the different species densities, so that the TME properties can be inferred from the measured densities. However, the dynamical system does not provide information about the morphology of the tumor, and its local composition. Indeed, this analysis poorly represents the structure of the tumor, which is divided into a nest, a cancer-associated stroma, and a healthy stroma.

For these reasons, in what follows we extend our model by incorporating the effects of cell species diffusion in order to obtain a more faithful spatio-dynamical description of the system, thus re-analyzing the cases above mentioned in this more general framework.

## Spatio-temporal behavior of tumor growth

Spatial study is an important tool in cancer diagnosis since the tumor shape reveals the aggressiveness of cancer cells and the role of their microenvironment (*Bearer et al., 2009*; *Cristini et al., 2005*; *Cristini et al., 2009*; *Huang et al., 2020*; *Balois and Ben Amar, 2014*; *Koay et al., 2018*; *Fraldi and Carotenuto, 2018*; *David and Perthame, 2021*; *Carotenuto et al., 2018*; *Perthame and Villa, 2024*). Similarly, the localization of immune cells in the tumor environment (*Mukherjee et al., 2023*), as well as proliferating and pre-metastatic cells often found in niches, are active areas of research in oncology and a valuable support for clinical prognosis (*Li and Neaves, 2006*; *Psaila and Lyden, 2009*; *Liu and Cao, 2016*).

In the previous section, we described our cell mixing through the global time evolution given by a dynamical system. Here, we aim to complete the modeling by considering the spatial heterogeneity of the interplay between fibroblasts, T-cells, and cancer cells and its consequences for tumor cell localization and proliferation. We first present the mixture model (*Balois and Ben Amar, 2014*; *Ackermann et al., 2021*), which is able to incorporate the spatial distribution and evolution of active cells leading to a set of partial differential equations that we solve with the finite element (FEM) software COMSOL Multiphysics (*Comsol Inc, 2024*). Last, we propose different mechanisms as sources of anisotropy in the tumor nest, including the introduction of a nematic tensor for the orientation of fibroblasts.

As in the previous section, we consider four different cell types namely cancer cells, T-cells, activated and non-activated fibroblasts, with different attraction properties. A component that does not play an active role in the mixture is also added via an inactive $N$ fraction. It concerns the intercellular fluid, the healthy cells, and the dead cells. This last component is also a source of material, for the proliferation of cancer cells. Each component is described by a local mass fraction $\phi_i$, a velocity $\mathbf{v}_i$, and a proliferation/death term $\Gamma_i$, with $\phi$ and $\Gamma$ denoting generic quantities that are now space and time-dependent. More specifically, $\phi_i$ corresponds to the local value of $C, T, F_A, F_{NA}$ introduced in the previous sections and $\phi_0 = 1 - \sum_{i \neq 0} \phi_i$ represents $N$ is the growth rate of each component, which is now space and time-dependent (see *Equations 1–4*) and may be positive only for cancer cells, since we consider that fibroblasts and T-cells are recruited from the surrounding environment. CAFs produce a significant amount of fibers, resulting in a higher friction between different species, chosen to be proportional to the local amount of CAFs. This friction, which is also a space-time-dependent quantity, will impact the dynamics of the mixture.

The sample also contains diffusive signaling molecules that are at the origin of the immune activity. In the previous section, chemicals were not introduced, although they were associated with some coefficients of the dynamical system. Here, the chemicals that determine the cell behavior, are represented by a concentration $c_j$ (*Olmeda and Ben Amar, 2019*; *Mori and Ben Amar, 2023*). Note that these chemicals have no mass and diffuse through the mixture with the diffusion coefficient $D_c$. For simplicity, we restrict ourselves to a single chemical of concentration $c$, that mediates both chemotaxis and activation of fibroblasts. The balance between its production and degradation rate writes $(\tau_{cC}^{-1} \phi_C - \delta_c)c$, since it is produced by cancer cells and naturally degraded. With these considerations, we study the case where the tumor is well supplied with nutrients, which are, therefore,

not explicitly mentioned, and we write a set of conservation equations for each component of the mixture:

$$\begin{cases} \partial_t \phi_i + \nabla.(\phi_i \mathbf{v}_i) = \Sigma_j \Gamma_{ij} \phi_j, \\ \partial_t c = D_c \nabla^2 c + \tau_{cC}^{-1} \phi_C - \delta_c c. \end{cases} \tag{8}$$

When integrated over the whole space and when the only boundary conditions for the velocity are considered, the first equation in *Equation 8* recovers the dynamical system. Besides, we consider that chemicals equilibrate much faster than the tissue dynamics, so that: $\lambda_c \nabla^2 c + \alpha_{cC} \phi_C - c = 0$, with $\lambda_c = \sqrt{D_c/\delta_c}$ the penetration length and $\alpha_{cC} = (\delta_c \tau_{cC})^{-1}$. We now present a derivation for the average local velocity $\mathbf{v}_i$ of each species by evaluating its momentum equation and the expressions for the various source terms.

## Momentum and free energy density derivation

Since the dynamics are very slow and completely controlled by dissipation, the Onsager variational principle of least dissipation (*Ackermann and Ben Amar, 2023*) yields the set of partial differential equations coupling the densities to the velocities in the mixture (*Ackermann et al., 2021*). This principle, introduced by Lord Rayleigh and further developed by Onsager (*Strutt, 1871*; *Onsager, 1931b*; *Onsager, 1931a*; *Onsager and Machlup, 1953*), is widely used in soft matter (*Doi, 2011*), in the biophysical context (*Ackermann et al., 2021*; *Balois and Ben Amar, 2014*; *Wang et al., 2021*; *Borja da Rocha et al., 2022*; *Ackermann and Ben Amar, 2023*) as well as other areas of physics (*Minguzzi, 2015*). First we define a free energy functional $\mathcal{F}$ from a free energy density $F$ integrated over the volume $V$, the associated chemical potentials $\mu_i$, and a dissipation function $\mathcal{W}$:

$$\begin{cases} \mathcal{F} = \int dV F(\{\phi_i\}, c) \quad ; \quad \mu_i = \frac{\delta \mathcal{F}}{\delta \phi_i} , \\ \mathcal{W} = \int \sum_{i \neq j} \frac{\xi_{ij} \phi_i \phi_j}{2} \left( \mathbf{v}_i - \mathbf{v}_j \right)^2 dV, \end{cases}$$

where the $\xi_{ij}$ are the relative friction coefficients between components $i$ and $j$, and $\mathbf{v_i}$ and $\mathbf{v_j}$ are the velocities of the different cell types. The Rayleighian is then defined as the sum of the dissipation function $\mathcal{W}$ and the rate of change of the free energy function $\mathcal{F}$. Within this framework, the final equations for the local velocities are obtained by minimizing the Rayleighian with respect to each velocity $\mathbf{v_i}$:

$$\mathcal{R} = \frac{d\mathcal{F}}{dt} + \mathcal{W}, \quad \frac{\delta \mathcal{R}}{\delta v_i} = 0 \Rightarrow \sum_j A_{ij} \mathbf{v}_j = -\nabla \mu_i ,$$

where $A_{ij}$ is the friction matrix and $\mu_i$ are the chemical potentials. Defining: $\tilde{\phi}_i = \phi_i \phi_0^{-1}$, the friction matrix reads:

$$\begin{cases} A_{ii} = \sum_{j \neq i, 0} (\xi_{ij} + \xi_{j0} \tilde{\phi}_i) \phi_j + \xi_{i0} (1 + \tilde{\phi}_i)^2 , \\ A_{ij} = \phi_j (-\xi_{ij} + \xi_{i0}(1 + \tilde{\phi}_i) + \xi_{j0}(1 + \tilde{\phi}_j) + \sum_{k \neq i,j,0} \xi_{k0} \tilde{\phi}_k). \end{cases}$$

Importantly, in our model, we assume that all frictions $\xi_{ij}$ are equal, except for the friction with the CAFs. In fact, the mass fraction of CAFs is assumed to reflect the amount of matrix produced by the fibroblasts, resulting in a very high friction in the medium. Therefore, we write: $\xi_{ij} = \xi_0, \xi_{iCAF} = \xi_0 + \xi_1 \phi_{CAF}$.

After deriving the momentum equations, we construct the free energy density. Building a free-energy density for a biological material is justified, because, although biological materials are out of equilibrium, their behavior often resembles that dictated by thermodynamics. It is, therefore, useful to write a free energy in terms of state variables. Following (*Ackermann et al., 2021*) and inspired by the Cahn-Hilliard approach, we define a free energy density $F$ as a sum of the interaction potential $f$ and of the cost induced by the mass fraction gradients $\kappa(\nabla \phi_i)^2/2$. We also assume for $f$, the Flory-Huggins free energy density of mixing that depends on the local mass fractions (or equivalently the volume fractions) of each component $\phi_i$ (*Flory, 1942*; *Huggins, 1941*):

$$\begin{cases} F = f + \sum_{i,j} \frac{\kappa}{2}(\nabla \phi_i)^2 \quad ; \quad \mu_i = \partial f/\partial \phi_i - \kappa \nabla^2 \phi_i , \\ f = \sum_i D_i \phi_i \log(\phi_i) - \sum_{i,j} \alpha_{ij} \phi_i \phi_j . \end{cases} \tag{9}$$

The terms $D_i \phi_i \log(\phi_i)$ control both the diffusion and volume exclusion, and the quadratic expansion $-\alpha_{ij}\phi_i\phi_j$ controls the attraction or repulsion between species.

In the following, we explain in more detail the effective free energy density of our system, as well as the expression of the different proliferation rates, and the values of the parameters, focusing on the attraction/repulsion and the chemotactic terms.

## Effective free energy density and source terms

We now apply the general formalism presented in the previous section to the cancer cell mixture using the same notation as Section Dynamic Modeling in the Lung Cancer TME for $C, T, F_{NA}, F_A$. However, while in the first section, these quantities represented averaged mass fractions over the whole tumor, here they are local mass fractions averaged over a small volume of the tumor but large compared to the cell size. We now detail the effective free energy, focusing first on the interaction potential $f$ defined in *Equation 9*.

The attraction between cancer cells is represented by $-\lambda_{CC}C^2$, between cancer cells and T-cells by $-\lambda_{CT}CT$, and between all types of fibroblasts and cancer cells by $-\lambda_{CF}C(F_{NA}+F_A)$, where the $\lambda$ coefficients are positive. These interactions are accounted for the free energy density that reads:

$$\begin{aligned} f = &D_C C \log(C) + D_T T \log(T) + D_{CAF} F_A \log(F_A) + D_{NA} F_{NA} \log(F_{NAF}) + D_0 \phi_0 \log(\phi_0) \\ &- \lambda_{CC} C^2 - \lambda_{CT} CT - \lambda_{CF} C(F_{NA} + F_A) \end{aligned} \tag{10}$$

Finally, we focus on the proliferation rates to complete the system *Equation 8*. Note that we must distinguish the cells produced in situ, such as the cancer cells $C$ and the activated fibroblasts $F_A$, from the cells attracted to the tumor nest ($T$ and $F_{NA}$) by the chemical $c$. We keep the growth rate of cancer cells $\Gamma_C$ given by Eq. 1 with $\delta_{TF} = 0$ because the inhibition of T-cells by fibroblasts is now treated by increasing the friction created by the fiber barrier. T-cells and NAFs do not proliferate and are not derived from precursors at the tumor site, but are instead generated far away from the tumor and are attracted to it by chemotaxis. Thus, at the boundaries $S$, their source rate is driven by an incoming flux due to the chemical $c$. For the NAFs, the volume term $\Gamma_{NA}|_V$ does not include the source term $\alpha_{NA,C}$ (see *Equation 3*) which is provided by the boundaries, and $F_{NA}$ is neglected:

$$\begin{cases} \Gamma_T|_S = \dfrac{c(1-S)}{\tau_T} \quad \text{and} \quad \Gamma_T|_V = -\delta TS, \\ \Gamma_{NA}|_S = \dfrac{c(1-S)}{\tau_F} \quad \text{and} \quad \Gamma_{NA}|_V = -K_A F_{NA} c - \delta F_{NA} S. \end{cases} \tag{11}$$

**Table 3.** Values of the parameters in the spatial model.

| Parameter | Value | Parameter | Value |
|---|---|---|---|
| $D_0$ | $10^6 \, kg.\mu m^{-1}.T_0^{-2}$ | $\delta$ | $1.18 \, T_0^{-1}$ |
| $D_C$ | $10^5 \, kg.\mu m^{-1}.T_0^{-2}$ | $\tau_T, \tau_{NA}$ | $2.5 \times 10^{-3} \, \mu m.T_0^{-1}$ |
| $D_T, D_{NAF}, D_{CAF}$ | $2.5 \times 10^5 \, kg.\mu m^{-1}.T_0^{-2}$ | $\delta_{CT}$ | $5 \, T_0^{-1}$ |
| $\lambda_{CC}$ | $2.5 \times 10^5 \, kg.\mu m^{-1}.T_0^{-2}$ | $\xi_0$ | $3 \times 10^2 \, kg.\mu m^{-3}.T_0^{-1}$ |
| $\lambda_{CT}$ | $7.5 \times 10^5 \, kg.\mu m^{-1}.T_0^{-2}$ | $\xi_1$ | $3 \times 10^5 \, kg.\mu m^{-3}.T_0^{-1}$ |
| $\lambda_{CF}$ | $0 \, kg.\mu m^{-1}.T_0^{-2}$ | $K$ | $10 \, T_0^{-1}.\mu m^2$ |
| $\kappa$ | $3.6 \times 10^7 \, kg.\mu m.T_0^{-2}$ | $\lambda_{cC}$ | $30 \mu m$ |

On the other hand, the production of CAFs is not changed and $\Gamma_A$ defined in *Equation 4* is still relevant, provided that the plasticity contribution is now proportional to the chemical concentration $c$ and not the cancer cells fraction.

We scale the various physical parameters in *Table 3*. Their values are derived from the literature and from the spatial structure of the TME, as well as the expected outcomes for the different scenarios, while the growth parameters are related to those in the first part of the article. For example, the tumor nest is a dense phase of cancer cells with mass fraction $C = 0.8$ that an interface of size $d \sim 10\,\mu m$ separates from the dilute phase $C \sim 0$. This choice, together with the typical value for the energy-density of $10^6\,kg.\mu m^{-1}.d^{-2}$ found in literature imposes the values of $D_C, D_0, \lambda_{CC}, \kappa$ (*Ackermann et al., 2021*). Similarly, we assume a weak infiltration of T-cells and fibroblasts into the nest, resulting in low values for the attraction parameters $\lambda_{CT}, \lambda_{CF}$ when compared to $\lambda_{CC}$. Since $D_{NAF}, D_{CAF}, D_T$ control the diffusion of the fibroblasts and T-cells, whose source is located at the boundaries, their values are determined by the density gradients between the boundaries and the tumor nest. At the same time, the times required for T-cells to kill the tumor nest is given by $\delta_{CT}$, and the times for the arrival of the T-cells and NAFs into the domain, as well as their average mass fractions, are governed by $\tau_T$ and $\tau_F$, respectively. The NAFs transform into CAFs in a time controlled by $K$. We use the letter $T_0$ as the time unit which corresponds to the typical division time for tumor development. For experiments in vitro in perfect conditions of nutrient access, the relevant time is the largest one, corresponding to the slowest process, so the typical division time of cancer cells. It is about 1 d and the same time is needed for the eradication of tumor cells (*Cellosaurus, 2023*; *Xie et al., 2010*; *Kloss et al., 2013*). But in vivo, it is obviously a very different time. It is more related to the time scale necessary for the tumor to grow in its natural environment and is of the order of months (*Winer-Muram et al., 2002*). The plasticity of fibroblasts is controlled by the chemical penetration length. Since we assume that only the fibroblasts lining the tumor nest are activated, we choose a low penetration length of $30\,\mu m$. At the same time, the ratio $\alpha$ between the chemical production and degradation parameters is set to $\alpha_{cC} = 3$.

In the following, we will show that, once activated by tumor cells, fibroblasts can inhibit tumor growth through a confinement effect but also limit the cytotoxic role of T-cells or simply prevent their infiltration into the tumor.

## Ambiguous role of fibroblasts in tumorigenesis

To illustrate the ambiguous role of fibroblasts in tumor progression, we explore different scenarios through a two-dimensional numerical study. This allows us to compare a fibrotic and a non-fibrotic tumor in the presence or absence of an immune response. Thus, the different growth cases we present are: a tumor without CAFs and T-cells, a tumor with both CAFs and T-cells, a tumor with T-cells but with a low level of CAFs, and a case with CAFs but no efficient T-cells. In the appendix, we present other scenarios of free growth: a TME in the presence of only NAFs and inefficient T-cells (*Figure 5—figure supplement 1B*), in presence of NAFs (*Figure 5—figure supplement 1C*), and a TME in which only cancer cells are present (*Figure 5—figure supplement 1D*). We first consider a single tumor nest before analyzing the case of two adjacent nests. Therefore, there is only one tumor nest at time $t = 0$ (*Figure 5A*).

As shown in *Figure 5E and H* (blue curve), in the case of a free-growing tumor, i.e., in the presence of NAFs and inactive T-cells, cancer cell growth is not hindered by any obstacles, so this is the most severe situation. However, the tumor provokes the formation of a stroma composed of the NAFs and inefficient T-cells. The presence of a stroma plays a role in the cancer cell growth described in *Equation 1* and leads to a growth that would be less important than in a case without stroma or composed only of NAFs (*Figure 5—figure supplement 1A and G*). In fact, the pressure exerted by the stroma increases the mass fraction of cancer cells in the core of the nest, as well as the total mass fraction $\mathcal{N}$ at the stroma-nest interface. This phenomenon corresponds to the competition for space and resources between the different species already described in the previous section. In the case of a free cancer growth without stroma, the tumor nest rapidly invades the environment and its surface fraction in the simulation window is almost 40% after reaching $t = 35\,T_0$, with a trend that is not yet saturated and an average mass fraction of cancer cells of 25%.

When fibroblasts are activated, they make the environment around the tumor nest fibrotic (*Figure 5B and C*), with a mass fraction around the nest reaching 20-30%. As explained earlier (in Section Momentum and Free Energy Density Derivation), the fibers are introduced in our model by

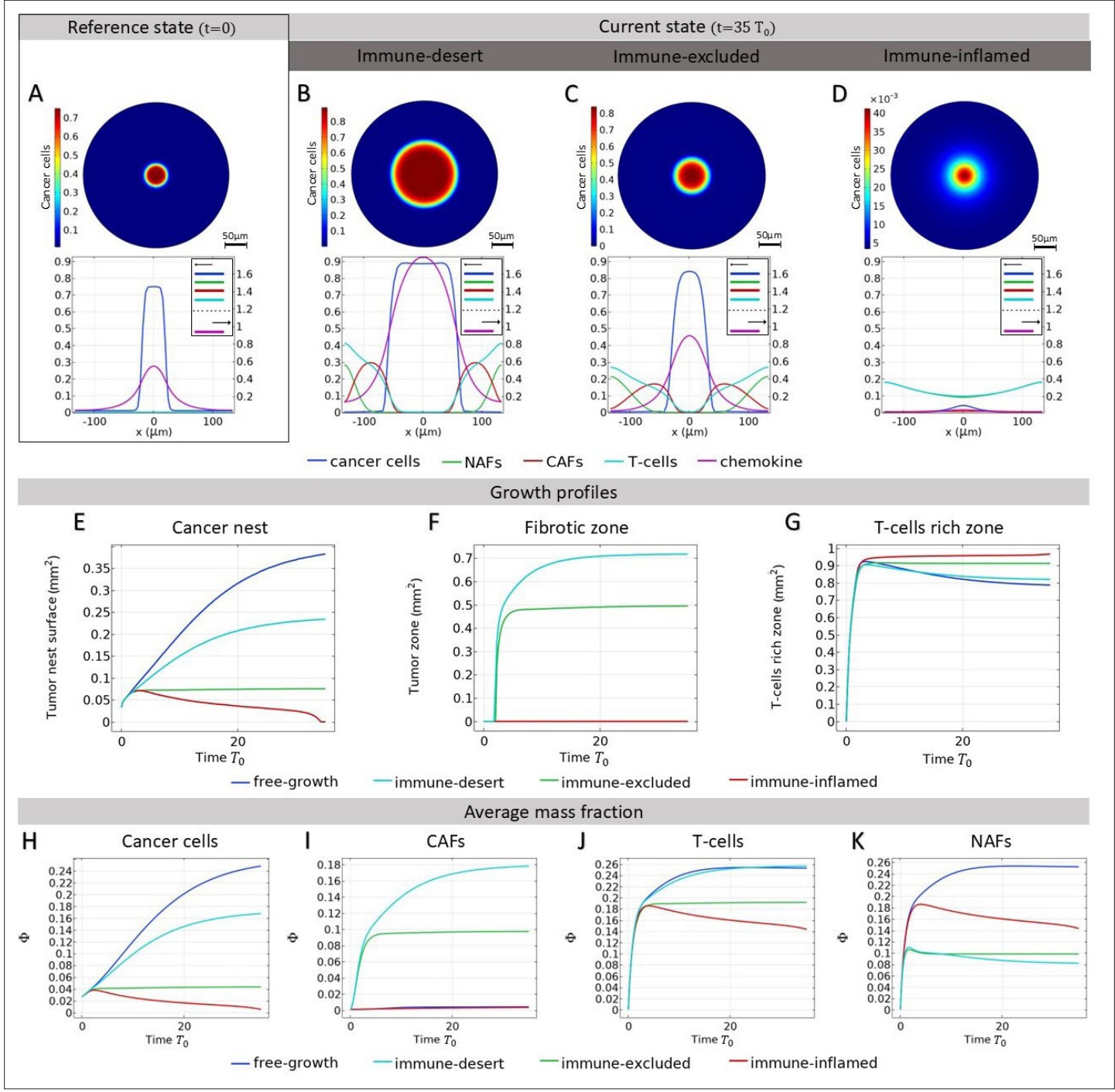

**Figure 5.** Small solid tumor growth. We refer here to the different tumor phenotypes described in *Figure 2*. (**A**) Mass fraction of cancer cells at time $t = 0$ and profile of different mass fractions on a section of the tumor. (**B-C-D**) Mass fraction of cancer cells at time $t = 35T_0$ and profile of mass fraction in immunodeficient tumor (**B**), when immune infiltration is inhibited by cancer-associated fibroblasts (CAFs) (**C**) and in immune-inflamed tumor (**D**) In **D**, the scale of the colorbar is $10^{-3}$ the values of (**A**), (**B**), (**C**), since this panel represents the case of efficient T-cells. (**E-F-G**) Development of different zones. (**E**) Surface fraction profile of the tumor nest for different scenarios, calculated as. $S/S_{total} = \int dV \delta(C > 0.1)/S_{total}$ (**F**) Fibrotic surface fraction profile for different scenarios, calculated as. $S/S_{total} = \int dV \delta(CAF > 0.1)/S_{total}$ (**G**) T-cell rich area fraction profile for different scenarios, calculated as $S = \int dV \delta(T > 0.1)/S_{total}$. (**H**) Cancer cell average mass fractions. (**I**) CAF average mass fractions. (**J**) T-cell average mass fractions. (**K**) Non-activated fibroblast (NAF) average mass fractions.

*Figure 5 continued on next page*

*Figure 5 continued*

The online version of this article includes the following figure supplement(s) for figure 5:

**Figure supplement 1.** Free growth of a tumor nest.

an increase of the friction coefficient between the different species and the activated fibroblasts, which is directly related to the fiber concentration. The tumor cells are then trapped behind a barrier with a very high friction which prevents the nest from expanding, and the surface area of the tumor nest decreases to 25% after $35\,T_0$, with a fibrotic area reaching 70%, while the average cancer cell decreases to 17% (*Figure 5E, F and G* cyan curve). The fibroblast population is composed of 30% NAFs and 70% CAFs, as the average fibroblast mass fraction is 26% (*Figure 5I and K* cyan curve).

At the same time, even in a situation where T-cells are efficient, the barrier precludes T-cells from the tumor (*Figure 5C*). In the latter case, tumor integrity is maintained and CAFs play a tumor-promoting role, inhibiting the immune response and stabilizing the tumor nest at 7.5% of the domain, with a fibrotic zone of 50% (*Figure 5E, F and H*, green curve). In this scenario, the average mass fractions are 5% for cancer cells, 10% for CAFs, 10% for NAFs (so 20% for the fibroblast population), and 20% for T-cells. It is interesting to note that this scenario quickly leads to a stable steady state. In contrast, the other simulations take longer times to reach a steady state. This could be due to the fact that the stroma builds up quickly compared to cell death when the cell population is not renewed, and that the steady state corresponds to a small nest that is reached after a short period of growth.

When T-cells are introduced without transformation from NAFs to CAFs (*Figure 5I*-red curve), the tumor dies as cancer cells are eliminated and the compact nest disappears (*Figure 5E*- red curve). As the stroma initially builds up due to the presence of a tumor, the fibroblast and T-cell populations slowly relax. However, the tumor stroma takes a long time to retract. This suggests that even if the cancer is cured, the effects on the tissue can be long-lasting. In this case of non-activation of NAFs, T-cells infiltrate the core of the nest thanks to the attraction of cancer cells (through the parameter $\lambda_{CT}$) and allows tumor reduction (*Figure 5D*). Therefore, their invasion is efficient only in the absence of fibers. (*Figure 5G*-red curve).

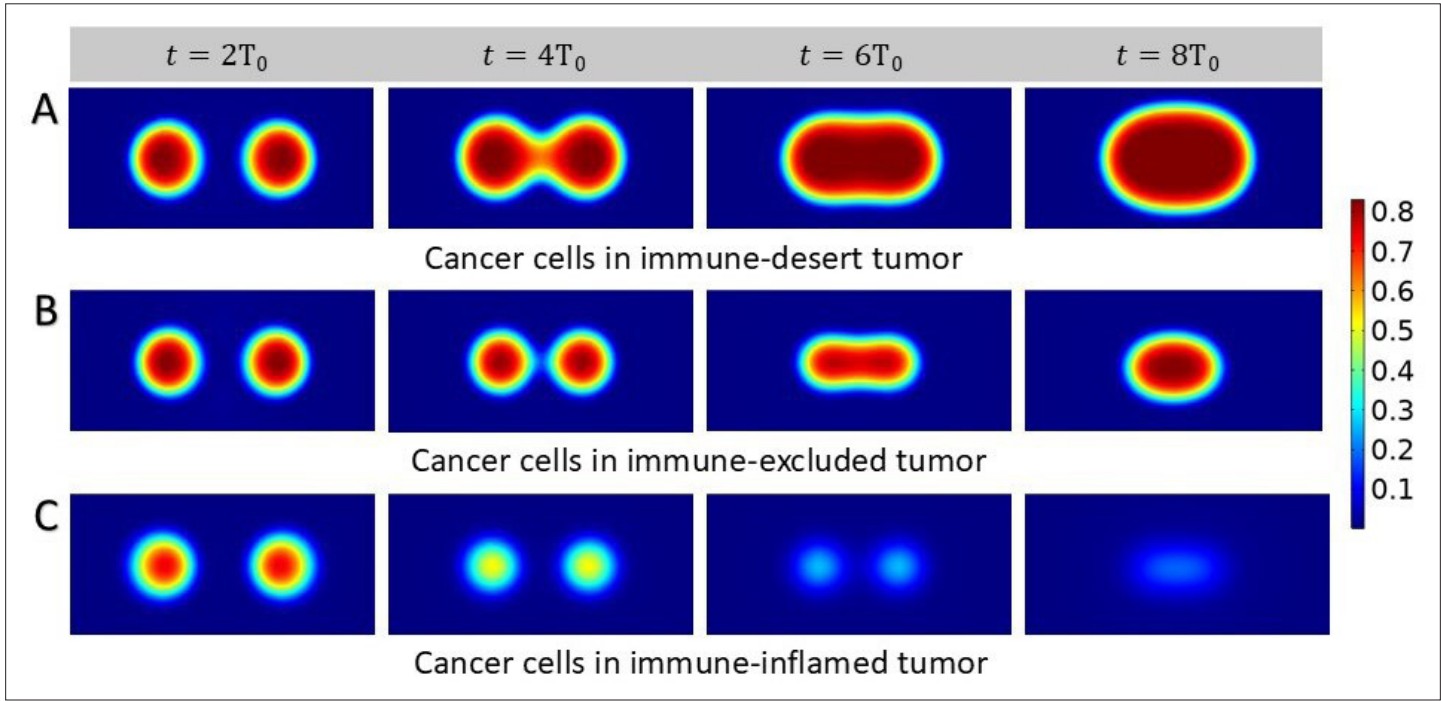

**Figure 6.** Growth process of two small solid tumors. Density plots showing cancer cell population. We start with two tumor nests placed at the same distance from the center of the domain (30 µm), sharing the same mass fraction of cancer cells and we present different scenarios. (**A**) Nests coalesce in immune-desert tumor, i.e., in the presence of inefficient T-cells. (**B**) Nests interacting in immune-excluded tumor, with chemotactic T-cells and cancer-associated fibroblasts (CAFs). (**C**) Nests coalesce in immune-inflamed tumor.

There are many mechanisms able to inhibit the immune system. In addition to the exclusion of T-cells from the tumor and the absence of active feedback, low attractiveness can also reduce the immune response. Indeed, when T-cells are less attracted, for example, because chemotaxis is not efficient enough, the tumor is free to expand. In *Figure 5B*, the cytotoxicity of T-cells is impaired, but their chemotaxis from the boundaries of the domains is not. This leads to an accumulation of inefficient T-cells around the tumor nest.

Next, we analyze the case where two tumor nests are nucleated and interact. We assume that they have the same initial size and mass fraction of cancer cells. In the absence of CAFs, an immune response is triggered and T-cell infiltration can occur. As expected, in the immune-inflamed tumor, the activity of T-cells reduces the ability of the two nests to coalesce by also reducing their mass fraction, leading to very low values of cancer cells (see *Figure 6C*). When T-cell activity is marginalized by the presence of CAFs, the two nests slowly coalesce to form a single solid tumor nest (see *Figure 6B*). In this case, growth is actually limited by the tumor-promoting function of CAFs. Although fibroblasts surround the tumor, its growth is enhanced due to the lack of immune cells inside. On the contrary, in the absence of T-cell chemotaxis, the tumor growth is unrestricted and the nests create a larger tumor by also increasing their mass fraction (see *Figure 6A*). In the latter cases, coalescence leads to anisotropic shapes of the tumor. Although relaxation in our model eventually leads to a round shape, this relaxation is slower for large tumors and even more for fibrotic tumors where friction significantly slows down this process. Patterns associated with coalescence may thus provide insights into the interpretation of anisotropic patterns in tumors. These are also related to the particular geometry of the system, such as the shape of the organ and the location of various blood vessels. In the next paragraph, we provide different directions to better describe this anisotropy.

## Toward anisotropy

To gain insight into the mechanisms leading to anisotropy, we numerically study different cases where the blood vessels are not spatially homogeneously distributed in the vicinity of the tumor nest. Thus, we introduce in our systems localized sources of T-cells and NAF, in the sense of *Figure 3*, where blood vessels are disk-like patterns in the tissue slice (*Figure 7A*), or sources corresponding to only a part of the boundaries, as it would be the case for a blood vessel located in the plane of the tissue slice (*Figure 7—figure supplement 1*). We note that the anisotropy in the sources can lead to non-trivial

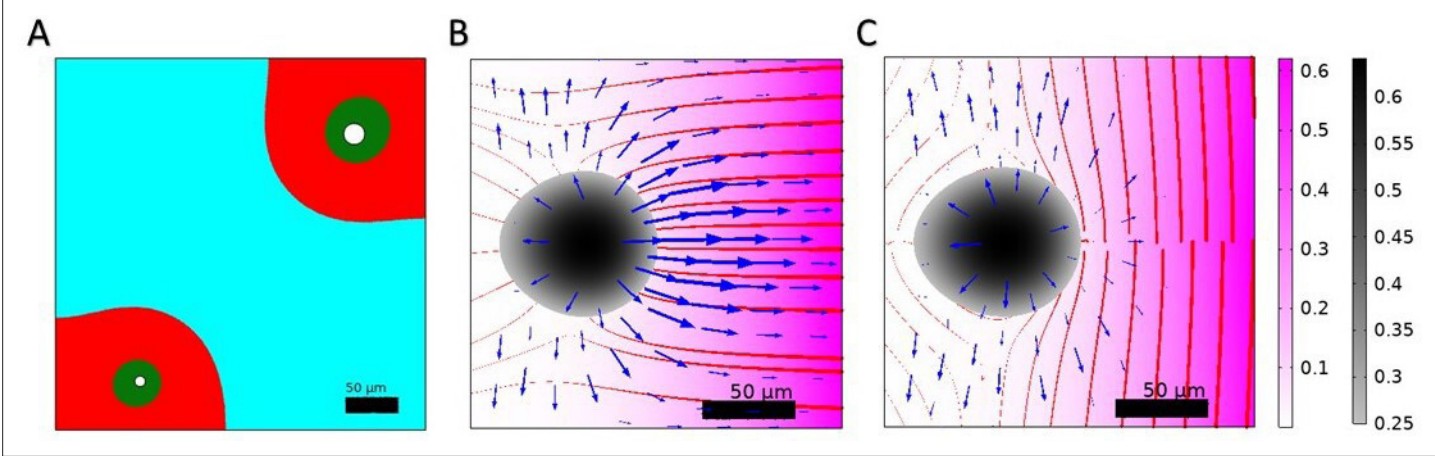

**Figure 7.** Anisotropy. Following biological observation reported in *Figure 3*, we here show different cases for describing anisotropy. (**A**) Example of the tumor microenvironment in the presence of two blood vessels (in white). The tumor nest is in cyan ($C > F_A + F_{NA} + T$), the cancerous stroma in red ($F_A > F_{NA} + T$) and the healthy stroma in green ($F_{NA} + T > F_A$). The tumor shape is dictated by the localization of the blood vessels from which T-cells and non-activated fibroblasts (NAFs) are issued. (**B-C**) Tumor behavior in the presence of normal (**B**) or orthonormal (**C**) fibers. The tumor fraction is indicated in colors from black to gray which we limit for $C > 0.25$ to better visualize of the contour of the nest. The flux $C\mathbf{v}_C$ is indicated with blue arrows whose thickness is proportional to the magnitude of the flux. The nematic order is shown with red lines whose thickness is proportional to the determinant of the matrix $\mathbf{Q}$ (see SI, Nematic order) and the fibroblast fraction is indicated with a pink color gradient.

The online version of this article includes the following figure supplement(s) for figure 7:

**Figure supplement 1.** Different tumor phenotypes in non-spherical domains.

shapes of the tumor tissue that are not due to instability but rather to the geometry of the system. These vessels are considered static, as dynamic vessels would require a more accurate modeling of angiogenesis. In the case of localized blood vessels modeled as disks, since the interface between the vessel and the system is smaller than in the case of boundary sources, the parameters driving T-cell population growth and T-cell efficiency in killing cancer cells must be increased in order to eradicate the tumor in the absence of cancer-associated fibroblasts. This highlights a peculiar phenomenon: increasing the density of blood vessels has ambiguous consequences. On the one hand, it allows a more efficient immune response and activate the arrival of T-cells. However, on the other hand, it also provides more nutrients to cancer cells and could ultimately favors metastasis. Therefore, critical processes such as the presence of a fibrous barrier around the tumor may have dual effects on and during tumor evolution. Furthermore, we introduce a more complete description of the fibrous stroma by adding a nematic order to the matrix and an anisotropic friction with respect to this matrix. A tensor order parameter $\mathbf{Q}$ is introduced which characterizes the average orientation of the fibroblasts and fibers and their degree of order (see Appendix 1 Section B, *Model for an anisotropic friction with the matrix*). Indeed, it has been shown that the density of the extracellular matrix is not always sufficient to induce T-cell exclusion (*Carstens et al., 2017*). We, therefore, provide the CAFs with an orientation, that is the result of there coupling with the matrix they deposit (*Li et al., 2017*; *Bell et al., 2025*). This long range order has been shown to induce an anisotropic friction (*Jacques et al., 2023*) that may be related to immune cell exclusion (*Sun et al., 2021*). At the same time, matrix orientation strongly influences metastasis and tumor growth by providing directional cues. Thus, there is a strong difference in the outcome of a tumor with fibers normal to the tumor surface and one with fibers along the tumor, both at the level of immune response and cancer escape. In *Figure 7B and C* we show the nematic order of the matrix and the flux field of the cancer cells in the case of orthonormal and normal orientation with respect to the tumor nest boundary, and due to the influx of fibroblasts from the right side of the system. The flux of cancer cells is much higher in the case of normal orientation of the fibers than in the case of parallel orientation. In conclusion, the description of T-cell exclusion by the sole density would be adapted to a case with fibers orientation aligned with the cancer nest surface, but not in the case with different orientations. Further investigations may, therefore, introduce these two elements to our model: a nematic order in the CAF layer, whose orientation is determined by coupling with the orientation of the interfaces, and an anisotropic friction, which is higher in the direction normal to the nematic matrix order than in the orientation of the matrix. Note that active stresses ($\boldsymbol{\sigma} = -\zeta\mathbf{Q}$ with $\zeta$ the activity) and couplings with the proliferation can develop in interaction with this nematic order, as explained and modeled in a previous article by some of us (*Ackermann and Ben Amar, 2023*), and a precise description of the nematic order may require to describe separately the fibroblasts and the matrix (*Li et al., 2017*; *Bell et al., 2025*; *Jacques et al., 2023*).

## Discussion

Our work provides a physical model to quantify the role of the immune system in human lung tumors. This is in line with different similar mathematical models, that study through this lens the inhibition/activation of the immune system by cancer cells either by the mean of nonlinear compartment models resembling our dynamical system, for instance regarding macrophage enrolment and cytokine signaling (*Arabameri et al., 2018*; *Li et al., 2019*), or mixture models (*Fotso et al., 2024*). We thus connect the two approaches in order to rigorously derive the parameters of the model and derive insights from both. In our article, the focus is on early tumor growth prior to angiogenesis and the development of its stroma rich in T-cells and fibroblasts, the activation of non-activated fibroblasts into cancer-associated fibroblasts and the marginalization of T-cells from the tumor nest. Other immune cells, such as macrophages, could complete this study in a further investigation.

After analyzing the data from the literature and from patients with LUSC or LUAD pathologies, (see *Table 1*), we propose a physical model, both theoretical and numerical, for the interactions between cancer cells, T-cells, and fibroblasts during tumor progression and their different roles. In particular, fibroblasts are introduced in the model in their inactive and active states (NAFs and CAFs). The modeling involves two steps: firstly, a dynamic study aimed at elucidating the key factors that can characterize the properties of the micro-environment on which the ability of T-cells to inhibit tumor growth depends. Scenarios scaled by only two parameters control the dynamics and evaluate the aggressiveness of the tumor. In particular, we have established the spatial organization of the

different cell types inside and outside the tumor core. The model is based on the continuous mixture model derived from Onsager's variational principle. Simulations were performed with the FEM software COMSOL Multiphysics in a two-dimensional framework. This provided a complete description of the tumor morphology and composition. In fact, we explored different scenarios to fully appreciate the role of all cell types involved. The results confirmed the experimental evidence that CAFs play a dual role in promoting and suppressing cancer cell proliferation. First, they alter the tumor microenvironment with a fiber barrier that reduces the motility and activity of T-cells, thereby promoting cancer cell growth. At the same time, this growth is limited and results to be lower than in the absence of active fibroblasts. Moreover, we have also shown that although the different parameters can take a wide range of values and continuously change the results of the numerical study, different scenarios can be drawn by looking only at the orders of magnitude.

In this study, we did not consider the fibers of the extracellular matrix as a component *per se*, but we considered them through the friction increase. We also provided the basics and started to describe the consequences of an anisotropic friction in relation to a nematic fiber orientation (see SI, Anisotropic friction). The same formalism could also integrate non trivial active stresses as done in the article (*Ackermann et al., 2021*).

Our physical model aims to limit the number of parameters as much as possible. This may seem far-fetched given the complexity of the biological system, especially in vivo. However, there are several reasons for this limitation. First, determining the range of values for these parameters is a complex task in the absence of direct measurement. The strategy we used in Section Dynamic Modeling in the Lung Cancer TME is to isolate each process corresponding to each parameter and thus construct the model step by step. In Section Spatio-Temporal Behavior of Tumor Growth we used both this latter strategy and the spatial structure of the mixing. In addition, the different combinations of the parameter values provide a wide variety of scenarios and it would be sufficient to modify these values to model new chemical entering the system. Limiting the complexity of the different processes allows us to focus on other aspects of the problem, such as the time evolution and the spatial structure, as well as a more precise quantitative study. In the same spirit, we numerically report in Section Spatio-Temporal Behavior of Tumor Growth a two-dimensional system without substrate, which yields results that we can consider similar as a three-dimensional system with an invariance in the third dimension.

Identifying the different scenarios that can occur in the TME and their likelihood is critical to predict the outcome of therapy (*Mori and Ben Amar, 2023*). Indeed, both pharmacodynamics and pharmacokinetics are highly dependent on the spatial structure and composition at the tumor site. In particular, the dynamic of arrival of the T-cell population is over simplified here via chemotaxis and discard auto-chemotaxis at the origin of a swarming displacement of an increased number of cytotoxic T-cells eventually, CAR-T cells (*Galeano Niño et al., 2020*). This mechanism, rare in the solid-tumor case, is sequential in time leading to a competition between killing efficiency and cancerous cell repair (*Weigelin et al., 2021*; *Weigelin and Friedl, 2022*) which will depend on the cancer type considered. Classification and quantitative characterization of the various outcomes can also help to monitor the treatment in real-time according to the evolution of the TME.

In conclusion, this numerical investigation may help to understand the limitations of the immune system in the face of solid tumor growth. Understanding quantitatively how immune cells are excluded from the tumor nest may be helpful in the drug design of T-cell-based therapies. Conversely, the numerical reproduction of various processes that can be observed in vivo is an important step in the context of personalized medicine. A next step in the theoretical and numerical investigation would, therefore, be to introduce drug molecules into the framework we have presented. Indeed extending our model to follow the tumor evolution under treatment and the associated internal dynamics from the early alveolar stage of the lung by incorporating the actual three-dimensional environment would also be a major step forward.

## Acknowledgements

MBA and HS acknowledge the financial support from ITMO Cancer of Aviesan within the framework of the 2021-2030 Cancer Control Strategy, on funds administrated by Inserm (PCSI 2021, MCMP 2022). JA acknowledges the financial support from ANR COLLAMOEBOID (ANR-20-CE13-0031). MF and CB acknowledge the financial support under the National Recovery and Resilience Plan (NRRP), Mission 4, Component 2, Investment 1.1, Call for tender No. 104 published on 2.2.2022 by the Italian

Ministry of University and Research (MUR), funded by the European Union – NextGenerationEU– Project Title 2022ATZCJN AMPHYBIA – CUP E53D23003040006 - Grant Assignment Decree No. 961 adopted on 30.06.2023 by the Italian Ministry of Ministry of University and Research (MUR). We thank Philippe Benaroch, Layla Mathieson, Xudong Xing for useful discussions.

## Additional information

### Funding

| Funder | Grant reference number | Author |
| --- | --- | --- |
| Inserm Transfert | ITMO-PCSI | Martine D Ben Amar |
| Agence Nationale de la Recherche | ANR-20-CE13-0031 | Joseph Ackermann |
| Ministry of University and Research | 2022ATZCJN AMPHYBIA – CUP E53D23003040006 | Massimiliano Fraldi Chiara Bernard |
| Inserm Transfert | MCMP 2022 | Martine D Ben Amar |

The funders had no role in study design, data collection and interpretation, or the decision to submit the work for publication.

### Author contributions

Joseph Ackermann, Conceptualization, Software, Formal analysis, Supervision, Investigation, Methodology, Writing – original draft, Writing – review and editing; Chiara Bernard, Software, Validation, Investigation, Writing – original draft, Writing – review and editing; Philemon Sirven, Helene Salmon, Data curation, Validation; Massimiliano Fraldi, Supervision, Validation, Investigation, Writing – review and editing; Martine D Ben Amar, Conceptualization, Formal analysis, Supervision, Funding acquisition, Investigation, Methodology, Writing – original draft, Project administration, Writing – review and editing

### Author ORCIDs

Joseph Ackermann ![ORCID] https://orcid.org/0000-0001-9218-6655
Martine D Ben Amar ![ORCID] https://orcid.org/0000-0001-9132-2053

Reviewer #1 (Public review): https://doi.org/10.7554/eLife.101885.3.sa1
Reviewer #2 (Public review): https://doi.org/10.7554/eLife.101885.3.sa2
Author response https://doi.org/10.7554/eLife.101885.3.sa3

## Additional files

### Supplementary files

MDAR checklist

### Data availability

Raw data used in *Table 1* are available in *Table 1—source data 1*. Comsol Multiphysics files used in this work are available on GitHub (copy archived at *Ackermann and Bernard, 2025*).

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

# Appendix 1

In this appendix, we show cases of free growth but in simple stroma: the case of a stroma composed by NAFs and inefficient T-cells (*Figure 5—figure supplement 1B*), a stroma composed by NAFs only (*Figure 5—figure supplement 1C*), and the case without stroma (*Figure 5—figure supplement 1D*), the tumor nest surface fraction in the simulation window for the different scenarios (8E), and the average mass fraction of the cancer cells and the NAFs (*Figure 5—figure supplement 1F, G*). The values of the parameters depending on the scenario ($\tau_T$ and $\tau_F$, *Equation 11*) are shown in *Appendix 1—table 1*.

**Appendix 1—table 1.** Parameter values varying depending on the scenario in *Figure 5—figure supplement 1*.

| Parameter | $\tau_T\,(\mu m.d^{-1})$ | $\tau_F\,(\mu m.d^{-1})$ |
|---|---|---|
| C-c+NAFs+inefficient T-cells | $2.5\times10^{-3}$ | $2.5\times10^{-3}$ |
| C-c+NAFs | $2.5\times10^{-6}$ | $2.5\times10^{-3}$ |
| C-c | $2.5\times10^{-6}$ | $2.5\times10^{-3}$ |

Besides, we show the evolution of the tumor in the case with efficient T-cells and NAFs without CAFs (*Figure 5—figure supplement 1H*).

We also obtain different configurations by assuming deformed domains in order to study the anisotropic response of the tumor stroma (*Figure 7—figure supplement 1A and B*).

In particular, the results obtained differ from those presented above by introducing T-cells and NAFs only from the upper side of the boundaries, this assuming the possibility that T-cells are able to reach the nest from different parts. Finally, different regions have been highlighted: in *light blue* the cancer cells nest, in *red* the barrier of activated fibroblasts, and in *green* the healthy stroma composed of NAFs and T-cells.

## Section A Dynamical system

In this section, we detail the calculations referred to in the dynamical system part of the article (Lung Cancer Microenvironment). More precisely, we demonstrate the scaling of the various parameters in the case where activated fibroblasts prevent T-cells from killing the cancer cells. With the found scaling, we show that the stationary solutions of this dynamical system are the only stable solutions.

First, we recall the complete system of equations describing the dynamics of the tumor microenvironment. $C, T, F_A, F_{NA}$ represent the cancer cells, T-cells, active fibroblasts, and inactive fibroblasts, respectively.

$$\begin{cases} \dfrac{dC}{dt} = C - \dfrac{\delta_{CT}CT}{1+\delta_{TF}F_A} - \delta_C C\mathcal{S}, \\ \dfrac{dT}{dt} = \alpha_{TC}C - \delta_T T\mathcal{S}, \\ \dfrac{dF_{NA}}{dt} = \alpha_{NA} + \alpha_{NA,C}C - K_A F_{NA}C - \delta_{NA}F_{NA}\mathcal{S}, \\ \dfrac{dF_A}{dt} = K_A F_{NA}C - \delta_A F_A \mathcal{S}, \end{cases} \tag{A2}$$

where $\delta_{CT}$ is the killing rate of cancer cells by T-cells in the absence of active fibroblasts, $\delta_{TF}$ is the inhibition constant of T-cells by the active fibroblasts, $\alpha_{TC}$ is the attraction rate of T-cells to the tumor, $\alpha_{NA}$ is the attraction rate of inactive fibroblasts in the healthy tissue, $\alpha_{NA,C}$ is the attraction rate of non-active fibroblasts in the presence of cancer cells, $K_A$ is the plasticity constant between active and non-active fibroblasts, and $\delta_C, \delta_T, \delta_{NA}, \delta_A$ are the death rates due to space and resource constraints of the different species. For simplicity, in this article, we consider $\delta_C = \delta_T = \delta_{NA} = \delta_A = \delta$. We also define the small parameter $\epsilon \ll 1$.

### Section A.1 Interaction between T-cells and cancer cells

We examine the interaction between T-cells and cancer cells in the nest in the absence of fibroblasts $F_{NA} = F_A = 0$ as described in the main text. This case would lead to the tumor eradication. We look for parameters that lead to significant growth inhibition such that $C \sim \epsilon$ at equilibrium.

The equilibrium equations read:

$$\begin{cases} \dfrac{dC}{dt} = 0 = C - \delta_{CT}CT - \delta C(C+T), \\ \dfrac{dT}{dt} = 0 = \alpha_{TC}C - \delta T(C+T). \end{cases} \tag{A3}$$

Three pairs of solutions exist:

$$\{C,T\} = \left\{ \begin{matrix} 0 \\ -\dfrac{\alpha_{TC}\delta^2 + 2\alpha_{TC}\delta\delta_{CT} - (\delta+\delta_{CT})\sqrt{\Delta} + \alpha_{TC}\delta_{CT}^2 + \delta^2 - \delta\delta_{CT}}{2\delta^2\delta_{CT}} \\ -\dfrac{\alpha_{TC}\delta^2 + 2\alpha_{TC}\delta\delta_{CT} + (\delta+\delta_{CT})\sqrt{\Delta} + \alpha_{TC}\delta_{CT}^2 + \delta^2 - \delta\delta_{CT}}{2\delta^2\delta_{CT}} \end{matrix} \right. , \begin{matrix} 0 \\ \dfrac{-\sqrt{\Delta} + \alpha_{TC}\delta + \alpha_{TC}\delta_{CT} + \delta}{2\delta\delta_{CT}} \\ \dfrac{\sqrt{\Delta} + \alpha_{TC}\delta + \alpha_{TC}\delta_{CT} + \delta}{2\delta\delta_{CT}} \end{matrix} \left. \right\}, \tag{A4}$$

with $\Delta = (\alpha_{TC}\delta + \alpha_{TC}\delta_{CT} + \delta)^2 - 4\alpha_{TC}\delta\delta_{CT}$.

Assuming $\delta_{CT} = \delta\epsilon^{-1}, \alpha_{TC} = a_0\epsilon$, with $\epsilon \ll 1, a_0 \sim 1$, the solutions, *Equation A4* simplify:

$$\{C,T\} = \left\{ \begin{matrix} 0 \\ \dfrac{\epsilon}{(a_0-1)\delta} \\ \dfrac{1-a_0}{\delta} + \mathcal{O}(\epsilon) \end{matrix} , \begin{matrix} 0 \\ \dfrac{\epsilon}{\delta} \\ \dfrac{a_0\epsilon}{\delta} \end{matrix} \right\}. \tag{A5}$$

The only solution leading to the tumor eradication is $\{C,T\} = \{\dfrac{\epsilon}{(a_0-1)\delta}, \dfrac{\epsilon}{\delta}\}$, and $a_0 > 1$.

We now check the linear stability of the different solutions by evaluating the Jacobian eigenvalues. A solution is stable if all its eigenvalues have a negative real part (*Strogatz, 2018*). The solution found is also the only stable solution provided that $a_0 > 1$. Here, the different eigenvalues of the Jacobian matrix are for the different solutions:

$$\{\omega_1,\omega_2\} = \left\{ \begin{matrix} 1 \\ -\dfrac{a_0\epsilon + \sqrt{\epsilon}\sqrt{a_0(2a_0-1)\epsilon - (a_0-1)^2}}{a_0-1} \\ \dfrac{a_0\epsilon}{a_0-1} + a_0 - 1 \end{matrix} , \begin{matrix} 0 \\ \dfrac{\sqrt{\epsilon}\sqrt{a_0(2a_0-1)\epsilon - (a_0-1)^2} - a_0\epsilon}{a_0-1} \\ a_0\left(\dfrac{a_0\epsilon}{a_0-1} + \epsilon + 1\right) - 1 \end{matrix} \right\}. \tag{A6}$$

The second solution with $a_0 > 1$ is the only stable solution.

## Section A.2 Role of activated fibroblasts on T-cells

We isolate a subsystem composed of cancer cells, T-cells, and active fibroblasts, as in Lung TME Composition, in order to determine the inhibition rate $\delta_{TF}$, responsible for the marginalization of the T-cells and subsequently for the increase of cancer cells at a fixed volume fraction of fibroblasts. We first assume that $F_A$ is constant. The equilibrium equations read:

$$\begin{cases} \dfrac{dC}{dt} = 0 = f_0 C - \Delta_{CT}CT - \delta C(C+T), \\ \dfrac{dT}{dt} = 0 = \alpha_{TC}C - \Delta_F T - \delta T(C+T), \end{cases}$$

where $f_0 = 1 - \delta F_A$, $\Delta_{CT} = \delta_{CT}(1 + \delta_{TF}F_A)^{-1}$ and $\Delta_F = \delta F_A$.

The different pairs of solutions read:

$$\{C,T\} = \left\{ \begin{array}{ccc} 0 & , & 0 \\ 0 & , & -\dfrac{\Delta f}{\delta} \\ \dfrac{\delta f_0 (\Delta_{CT} - \delta) - (\Delta_{CT} + \delta)\left(\alpha_{CT}(\Delta_{CT} + \delta) + \delta\Delta f - \sqrt{(\alpha_{CT}(\Delta_{CT} + \delta) + \delta\Delta f + \delta f_0)^2 - 4\delta f_0 \alpha_{CT}\Delta_{CT}}\right)}{2\delta^2 \Delta_{CT}} & , & \\ & & \dfrac{-\sqrt{(\alpha_{CT}(\Delta_{CT} + \delta) + \delta\Delta f + \delta f_0)^2 - 4\delta f_0 \alpha_{CT}\Delta_{CT}} + \delta\alpha_{CT} + \alpha_{CT}\Delta_{CT} + \delta\Delta f + \delta f_0}{2\delta\Delta_{CT}} \\ \dfrac{\delta f_0 (\Delta_{CT} - \delta) - (\Delta_{CT} + \delta)\left(\alpha_{CT}(\Delta_{CT} + \delta) + \delta\Delta f + \sqrt{(\alpha_{CT}(\Delta_{CT} + \delta) + \delta\Delta f + \delta f_0)^2 - 4\delta f_0 \alpha_{CT}\Delta_{CT}}\right)}{2\delta^2 \Delta_{CT}} & , & \\ & & \dfrac{\sqrt{(\alpha_{CT}(\Delta_{CT} + \delta) + \delta\Delta f + \delta f_0)^2 - 4\delta f_0 \alpha_{CT}\Delta_{CT}} + \delta\alpha_{CT} + \alpha_{CT}\Delta_{CT} + \delta\Delta f + \delta f_0}{2\delta\Delta_{CT}} \end{array} \right\}.$$

(A7)

We assume $\delta_{CT} = \delta\epsilon^{-1}, \alpha_{TC} = a_0\epsilon, \delta_{TF} = d_0\delta\epsilon^{-1}$, as justified in the main text. We obtain the following solution pairs:

$$\{C,T\} = \left\{ \begin{array}{ccc} 0 & , & 0 \\ 0 & , & -\dfrac{f_a}{\delta} \\ \dfrac{f_0}{\delta} + \mathcal{O}(\epsilon) & , & \dfrac{a_0 f_0}{\delta(f_0 + f_a)}\epsilon \\ -\dfrac{f_a(d_0 f_0 + d_0 f_a + 1)}{\delta} + \mathcal{O}(\epsilon) & , & \dfrac{d_0 f_a(f_0 + f_a)}{\delta} + \mathcal{O}(\epsilon) \end{array} \right\}.$$

(A8)

The only physical solution is the third pair: $\{C,T\} = \left\{ \dfrac{f_0}{\delta} + \mathcal{O}(\epsilon), \dfrac{a_0 f_0}{\delta(f_0 + f_a)}\epsilon \right\}$.

The eigenvalues of the Jacobian matrix read:

$$\{\omega_1, \omega_2\} = \left\{ \begin{array}{ccc} -f_a & , & f_0 \\ f_a & , & \left(\dfrac{1}{d_0} + f_0 + f_a\right) \\ -(f_0 + f_a) & , & -f_0 \\ \dfrac{f_a - i\sqrt{f_a}\sqrt{4f_0(d_0 f_0 + 2d_0 f_a + 1) + f_a(4d_0 f_a + 3)}}{2} & , & \dfrac{f_a + i\sqrt{f_a}\sqrt{4f_0(d_0 f_0 + 2d_0 f_a + 1) + f_a(4d_0 f_a + 3)}}{2} \end{array} \right\},$$

(A9) and

the third solution is the only stable one.

## Section A.3 Fibroblast plasticity

We consider the system described in the main text in the subsection *Fibroblast plasticity*, where the non-active fibroblasts population is maintained at a fraction $F_{NA} = f_n \delta^{-1}$ where $f_n < 1$ is a constant.

We write the equilibrium equations, where we replace $\delta_{CT} = \delta\epsilon^{-1}, \alpha_{TC} = a_0\epsilon, \delta_{TF} = d_0\delta\epsilon^{-1}$:

$$\begin{cases} \dfrac{dC}{dt} = 0 = C - \dfrac{\delta\epsilon^{-1}CT}{1 + d_0\delta\epsilon^{-1}F_A} - \delta C\mathcal{S}, \\ \dfrac{dT}{dt} = 0 = a_0\epsilon C - \delta T\mathcal{S}, \\ \dfrac{dF_A}{dt} = 0 = K_A f_n C/\delta - \delta F_A \mathcal{S}, \end{cases}$$

(A10)

where $\mathcal{S} = C + T + F_A + F_{NA}$.

Equating $C = \delta T\mathcal{S}/(a_0\epsilon)$ and $C = \delta^2 F_A\mathcal{S}/(K_A f_n)$ leads to $F_A = K_A f_n T/(\delta a_0\epsilon)$. The system rewrites:

$$\begin{cases} 0 = (1 - f_n)C - \left(\dfrac{\delta\epsilon^{-1}}{1 + d_0\epsilon^{-2}K_A f_n T/a_0} + \dfrac{K_A f_n}{a_0\epsilon}\right)CT - \delta C(C + T), \\ 0 = a_0\epsilon C - f_n T - \dfrac{K_A f_n}{a_0\epsilon}T^2 - \delta T(C + T). \end{cases}$$

(A11)

Since the active fibroblasts should prevent T-cells to kill cancer cells, but without reducing the T-cell population, we write $C \sim \delta^{-1}$ and $T \sim \epsilon\delta^{-1}$. In addition, the term related to the killing of cancer-cells by T-cells should be of order $\epsilon$. This leads to: $K_a \sim \delta$.

We now look for the solutions of the dynamical system, as expansions in $\epsilon$. Obvious solutions are: $C = 0, T = 0$ and $C = 0, T = -f_n\epsilon/(\delta\epsilon + K_A f_n/a_0)$. We look for the other solutions in the form:

$C = c_0 + c_1\epsilon, T = t_0 + t_1\epsilon$. The order $1/\epsilon$ in the dynamical system lead to $t_0 = 0$. The orders 1 and $\epsilon$ in the dynamical system leads to:

$$0 = (1 - f_n) - \frac{K_A f_n}{a_0} t_1 - \delta c_0$$
$$0 = a_0 c_0 - f_n t_1 - \frac{K_A f_n}{a_0} t_1^2 - \delta t_1 c_0 \tag{A12}$$

Finally the different solutions are:

$$\{C, T\} = \begin{cases} 0 & , & 0 \\ 0 & , & -\frac{a_0\epsilon}{K_a} + \mathcal{O}(\epsilon^2) \\ \frac{1 - f_n}{K_a f_n + \delta} + \mathcal{O}(\epsilon) & , & \frac{a_0\epsilon(1 - f_n)}{K_a f_n + \delta} + \mathcal{O}(\epsilon^2) \end{cases} \tag{A13}$$

with only one physical solution apart from the disappearance of both T-cells and cancer cells.

We now check the stability of the different solutions. The eigenvalues for the growth rates are:

$$\{\omega_1, \omega_2\} = \begin{cases} 1 - f_n & , & -\epsilon f_n \\ 1 & , & \epsilon f_n \\ -\frac{\epsilon(K_a f_n + \delta)}{\delta} & , & -\frac{\delta(1 - f_n)}{K_a f_n + \delta} + \mathcal{O}(\epsilon) \end{cases} \tag{A14}$$

and the solution $C = \frac{1 - f_n}{K_a f_n + \delta}, T = \frac{a_0\epsilon(1 - f_n)}{K_a f_n + \delta}, F_A = \frac{K_A f_n(1 - f_n)}{\delta(K_a f_n + \delta)}$ is the only stable solution.

## Section A.4 Non active fibroblasts attraction to tumor

Last, we demonstrate the results of the subsection *Tumor fibroblast attraction* of the main text. In order to analyze non active fibroblasts to tumor, we now consider the full dynamical system:

$$\begin{cases} \frac{dC}{dt} = 0 = C - \frac{\delta\epsilon^{-1}CT}{1 + d_0\delta\epsilon^{-1}F_A} - \delta C\mathcal{S}, \\ \frac{dT}{dt} = 0 = a_0\epsilon C - \delta T\mathcal{S}, \\ \frac{dF_{NA}}{dt} = 0 = \delta\epsilon^4 + \alpha_{NA,C}C - k\delta F_{NA}C - \delta F_{NA}\mathcal{S}, \\ \frac{dF_A}{dt} = 0 = k\delta F_{NA}C - \delta F_A\mathcal{S}. \end{cases} \tag{A15}$$

We look for solutions of the type: $\{C, T, F_{NA}, F_A\} = \delta^{-1}\{c_0 + \mathcal{O}(\epsilon), t_0\epsilon + \mathcal{O}(\epsilon^2), f_{NA} + \mathcal{O}(\epsilon), f_A + \mathcal{O}(\epsilon)\}$ for the case of fibroblasts preventing T-cells getting access to the tumor.

The dynamical system becomes to the first order in $\epsilon$:

$$\begin{cases} 0 = 1 - \frac{t_0}{1 + d_0\epsilon^{-1}f_A} - (c_0 + t_0\epsilon + f_{NA} + f_A), \\ 0 = a_0 c_0 - t_0(c_0 + t_0\epsilon + f_{NA} + f_A), \\ 0 = \alpha_{NA,C}c_0 - kf_{NA}c_0 - f_{NA}(c_0 + t_0\epsilon + f_{NA} + f_A), \\ 0 = kf_{NA}c_0 - f_A(c_0 + t_0\epsilon + f_{NA} + f_A). \end{cases} \tag{A16}$$

In order for $f_A, f_{NA}$ to be of order 0 in $\epsilon$, $\alpha_{NA,C}$ must also be of order 0.

At the lowest order in $\epsilon$ we obtain:

$$\begin{cases} 0 = 1 - (c_0 + f_{NA} + f_A), \\ 0 = a_0 c_0 - t_0(c_0 + f_{NA} + f_A), \\ 0 = \alpha_{NA,C}c_0 - kf_{NA}c_0 - f_{NA}(c_0 + f_{NA} + f_A), \\ 0 = kf_{NA}c_0 - f_A(c_0 + f_{NA} + f_A). \end{cases} \tag{A17}$$

The solutions of the full system write:

$$\{C, T, F_A, F_{NA}\} = \left\{ \begin{array}{c} 0 \\ \dfrac{1}{\alpha_{NA,C}+1} \end{array} , \begin{array}{c} t\epsilon \\ \dfrac{a_0}{\alpha_{NA,C}+1} \end{array} , \begin{array}{c} -f_{NA} \\ \dfrac{\alpha_{NA,C}k}{(\alpha_{NA,C}+1)(\alpha_{NA,C}+k+1)} \end{array} , \begin{array}{c} f_{NA} \\ \dfrac{\alpha_{NA,C}}{\alpha_{NA,C}+k+1} \end{array} \right\}.$$

(A18)

Only the second solution is physical.

We now check the stability of those solutions. The linear stability analysis yields the eigenvalues for the stability rates:

$$\{\omega_1, \omega_2, \omega_3, \omega_4\} = \left\{ \begin{array}{c} 1 \\ -1 \end{array} , \begin{array}{c} 0 \\ -1 \end{array} , \begin{array}{c} 0 \\ -\epsilon \end{array} , \begin{array}{c} 0 \\ -\dfrac{\alpha_{NA,C}+k+1}{\alpha_{NA,C}+1} \end{array} \right\}.$$

(A19)

The only stable solution is the second solution where all cell fractions are above 0 and below 1.

## Section B Model for an anisotropic friction with the matrix

### Section B.1 Nematic order

We introduce a refined model for the fiber role in the tumor microenvironment. We first add a nematic order to the CAF population since its structure involves a more or less disordered nets of fibers. This nematic order is described by a traceless nematic tensor $\boldsymbol{Q}$. We recall that the nematic tensor may be written as:

$$\boldsymbol{Q} = \frac{S}{2} \begin{pmatrix} \cos(2\theta) & \sin(2\theta) \\ \sin(2\theta) & -\cos(2\theta) \end{pmatrix}$$

(B20)

where $S$ is the amplitude of the anisotropy, and $\theta$ the angle for the nematic orientation. Besides, $\boldsymbol{Q}$ is determined by the minimization of the following free-energy:

$$\mathcal{F}_{nem} = \int d\Omega \left\{ \alpha \boldsymbol{Q} : \boldsymbol{Q} + \beta(\boldsymbol{Q} : \boldsymbol{Q})^2 + \frac{K}{2}(\nabla \boldsymbol{Q})^2 + \gamma \boldsymbol{Q} : \boldsymbol{\Phi} \right\}$$

(B21)

where $\alpha = -F_A$, and $\boldsymbol{\Phi}$ indicates the orientation of the tumor nest boundary: $\Phi_{ij} = \frac{1}{2}((\partial_i C)(\partial_j C) - 2(\partial_k C)^2)$. The orientation of the fibers with regards to the tumor nest is controlled by the parameter $\gamma$. In the spirit of our work, we assume that fibers orient fast compared to the growth times, so that we consider the following equation:

$\dfrac{\delta \mathcal{F}_{nem}}{\delta \boldsymbol{Q}} = 0$ (B22) which allows to evaluate $\boldsymbol{Q}$.

### Section B.2 Anisotropic friction

We consider an anisotropic friction between the different components and the CAFs in the dissipation function.

$$\mathcal{W}_{anis} = \int d\Omega \left\{ \frac{\xi_\parallel}{2}(\mathbf{v}_X^\parallel - \mathbf{v}_{FA}^\parallel)^2 + \frac{\xi_\perp}{2}(\mathbf{v}_X^\perp - \mathbf{v}_{FA}^\perp)^2 \right\}$$

(B23)

where $\mathbf{v}_X^\parallel$ is the velocity component along the unit vector of the nematic order orientation of the matrix while $X$ represents one component of the mixture $(C, F_A, F_{NA}, T)$. In the same way, $\mathbf{v}_X^\perp = \mathbf{v}_X - \mathbf{v}_X^\parallel$. after some algebra, this leads to the nematic friction tensor that depends only on the tensor $\boldsymbol{Q}$. More justifications can be found in *Jacques et al., 2023*.

## Section C Raw data for *Table 1*

The raw data for the two last lines of *Table 1* in the main text is provided in *Table 1—source data 1*.

The tumor stroma ratio is computed as follows:

$$TSR = \frac{\text{Tumor area}}{\text{Stroma area}}$$

The total surface fraction of cancer cells in the tumor is:

$$\phi_C = \frac{\text{Tumor area}}{\text{Stroma area} + \text{Tumor area}}$$

The surface fraction of fibroblasts in the stroma is:

$$\phi_f = \frac{\text{ADH1B} + \text{FAP} + \text{aSMA} - (\text{FAP} + \text{aSMA})}{\text{Stroma area}}$$

The data obtained for the number of CD3$^+$ and CD8$^+$ T-cells both in the stroma and in the tumor islands suggest that their surface fraction is much lower than the surface fractions of cancer cells or fibroblasts. The number of CD3$^+$ cells obtained is assumed to represent the total population of mature T-cells. If we multiply the number of these subsets of T-cells by the estimated area of each T-cell ($\pi R^2$, with R~5 μm) and compare it with the area of the tumor nest and the total area of fibroblasts, we find that the surface fraction is very low (1.4±2.6%) for LUAD and 0.3±0.4% for LUSC. Therefore, in most of this article, we considered a low surface fraction of T-cells compared to those of cancer cells and fibroblasts.

