## [Editor Report · eLife Assessment]

This is a **valuable** report of a spatially-extended model to study the complex interactions between immune cells, fibroblasts, and cancer cells, providing insights into how fibroblast activation can influence tumor progression. The model opens up new possibilities for studying fibroblast-driven effects in diverse settings, which is crucial for understanding potential tumor microenvironment manipulations that could enhance immunotherapy efficacy. While the results presented are **convincing** and follow logically from the model's assumptions, some of these assumptions, as acknowledged by the authors, may oversimplify certain aspects in light of complex experimental findings, system geometry, and general principles of active matter research. Nonetheless, the authors provide justification for their work as a meaningful step towards more comprehensive modeling approaches.

---

## [Referee Report · Reviewer #1 (Public review)]

The authors present an important work where they model some of the complex interactions between immune cells, fibroblasts and cancer cells. The model takes into account the increased ECM production of cancer-associated fibroblasts. These fibres trap the cancer but also protect it from immune system cells. In this way, these fibroblasts' actions both promote and hinder cancer growth. By exploring different scenarios, the authors can model different cancer fates depending on the parameters regulating cancer cells, immune system cells and fibroblasts. In this way, the model explores non-trivial scenarios. An important weakness of this study is that, though it is inspired by NSCLC tumors, it is still far from modelling tumor lesions with morphologies similar to NSCLC tumors and does not explore the formation of ramified tumors. In this way, is a general model and it is challenging how it can be adapted to simulate more realistic tumor morphologies.

Comments on revisions:

The authors have improved the manuscript and addressed my concerns.

---

## [Referee Report · Reviewer #2 (Public review)]

Summary:

The authors develop a computational model (and a simplified version thereof) to treat an extremely important issue regarding tumor growth. Specifically, it has been argued that fibroblasts have the ability to support tumor growth by creating physical conditions in the tumor microenvironment that prevent the relevant immune cells from entering into contact with, and ultimately killing, the cancer cells. This inhibition is referred to as immune exclusion. The computational approach follows standard procedures in the formulation of models for mixtures of different material species, adapted to the problem at hand by making a variety of assumptions as to the activity of different types of fibroblasts, namely "normal" versus "cancer-associated". The model itself is relatively complex, but the authors do a convincing job of analyzing possible behaviors and attempting to relate these to experimental observations.

Strengths:

As mentioned, the authors do an excellent job of analyzing the behavior of their model both in its full form (which includes spatial variation of the concentrations of the different cellular species) and in its simplified mean field form. The model itself is formulated based on established physical principles, although the extent to which some of these principles apply to active biological systems is perhaps debatable (see Weaknesses). The results of the model do indeed offer some significant insights into the critical factors which determine how fibroblasts might affect tumor growth; these insights could lead to new experimental ways of unraveling these complex sets of issues and enhancing immunotherapy. In this revised version, the authors have properly placed this work within the general context of other research on modeling the tumor-immune ecology.

Weaknesses:

Models of the form being studied here rely on a large number of assumptions regarding cellular behavior. One major issue is the degree to which close-to-equilibrium assumptions (such as the dynamics being driven by free energy minimization) can be taken as reliable predictors of the obviously active dynamics of biological cells. The authors have recognized this conceptual issue and have argued that these assumptions provide a reasonable first step for understanding the full complexity of dynamics in the tumor microenvironment.

The problem of T cell infiltration as well as the patterning of the extracellular matrix (ECM) by fibroblasts necessarily involve understanding cell proliferation, cell motion and cell interactions due e.g. to cell signaling. There is evidence that inherently non-equilibrium interactions between the fibroblasts and the extracellular matrix can lead to patterning of the fiber network and trapping of potentially infiltrating T-cells. it is not clear the extent to which this type of interaction can be captured by the approach being used here, although the authors propose that they can be mimicked by proper terms in their formulation. This to me is the primary concern that I had with this paper.

The authors have now addressed what used to be a separate weakness concerning the assumption that fibroblasts affect T cell behavior primarily by just making a more dense ECM. Instead, the organization of the ECM (for example, its anisotropy) could be playing a much more essential role than is given credit for here. This possibility is now discussed in some detail and the authors have suggested that the introduction of a nematic order parameter field would be a useful way to treat this effect.

---

## [Author Response]

The following is the authors’ response to the original reviews.

**eLife Assessment**
This is a useful report of a spatially-extended model to study the complex interactions between immune cells, fibroblasts, and cancer cells, providing insights into how fibroblast activation can influence tumor progression. The model opens up new possibilities for studying fibroblast-driven effects in diverse settings, which is crucial for understanding potential tumor microenvironment manipulations that could enhance immunotherapy efficacy. While the results presented are solid and follow logically from the model’s assumptions, some of these assumptions may require further validation, as they appear to oversimplify certain aspects in light of complex experimental findings, system geometry, and general principles of active matter research.

We thank the editor for recognizing the usefulness of our work. This work does not aim to precisely describe the complexity of the tumor microenvironment in lung cancer, but rather to classify and rigorously calibrate a minimum number of parameters to the clinical data we collect and generate, and reproduce the global structures of the microenvironment. We identify different scenarios, and show how they depend on the local interactions within this framework. Although we started in the first version with coalescence in the main text and anisotropic geometry in the supporting information, we realized that we needed to provide more directions to better show how our model can be extended. Thus, we added an analysis of a microenvironment with blood vessels, and showed how to introduce anisotropic friction as a function of fiber orientation, as well as active stress, paving the way for further studies, that would make our model more complex. However, in a first step, it is crucial to start with a limited number of parameters that can be rigorously determined, and this is how this first work was conceived.

**Public Reviews:**

**Reviewer #1 (Public review):**
The authors present an important work where they model some of the complex interactions between immune cells, fibroblasts and cancer cells. The model takes into account the increased ECM production of cancer-associated fibroblasts. These fibres trap the cancer but also protect it from immune system cells. In this way, these fibroblasts’ actions both promote and hinder cancer growth. By exploring different scenarios, the authors can model different cancer fates depending on the parameters regulating cancer cells, immune system cells and fibroblasts. In this way, the model explores non-trivial scenarios. An important weakness of this study is that, though it is inspired by NSCLC tumors, it is restricted to modelling circular tumor lesions and does not explore the formation of ramified tumors, as in NSCLC. In this way, is only a general model and it is not clear how it can be adapted to simulate more realistic tumor morphologies.

We thank the reviewer for highlighting the importance of our work. We acknowledge that although we provided anisotropic geometries and the study of the coalescence in the first version, more effort was needed to provide tools to extend our formalism to non-ideal cases. This is now added as Section ‘Toward anisotropy’, where we analyze the impact of blood vessels, and the anisotropic friction due to the nematic order for the fibers; this nematic order can also be used to introduce active nematic stress.

**Reviewer #2 (Public review):**
Summary:The authors develop a computational model (and a simplified version thereof) to treat an extremely important issue regarding tumor growth. Specifically, it has been argued that fibroblasts have the ability to support tumor growth by creating physical conditions in the tumor microenvironment that prevent the relevant immune cells from entering into contact with, and ultimately killing, the cancer cells. This inhibition is referred to as immune exclusion. The computational approach follows standard procedures in the formulation of models for mixtures of different material species, adapted to the problem at hand by making a variety of assumptions as to the activity of different types of fibroblasts, namely “normal” versus “cancer-associated”. The model itself is relatively complex, but the authors do a convincing job of analyzing possible behaviors and attempting to relate these to experimental observations.Strengths:As mentioned, the authors do an excellent job of analyzing the behavior of their model both in its full form (which includes spatial variation of the concentrations of the different cellular species) and in its simplified mean field form. The model itself is formulated based on established physical principles, although the extent to which some of these principles apply to active biological systems is not clear (see Weaknesses). The results of the model do offer some significant insights into the critical factors which determine how fibroblasts might affect tumor growth; these insights could lead to new experimental ways of unraveling these complex sets of issues and enhancing immunotherapy.

We thank the referee for this summary and for recognizing the strengths of our paper.

Weaknesses:Models of the form being studied here rely on a large number of assumptions regarding cellular behavior. Some of these seemed questionable, based on what we have learned about active systems. The problem of T cell infiltration as well as the patterning of the extracellular matrix (ECM) by fibroblasts necessarily involve understanding cell motion and cell interactions due e.g. to cell signaling. Adopting an approach based purely on physical systems driven by free energies alone does not consider the special role that active processes can play, both in motility itself and in the type of self-organization that can occur due to these cell-cell interactions. This to me is the primary weakness of this paper.

We thank the referee for this important comment, that allows us to clarify this important point. Although biological materials are out of equilibrium, their behavior often resembles that dictated by thermodynamics. Hence the usefulness of constructing a free energy, in terms of these variables. In a first approach to decipher the complex interactions and describe the different and sometimes non-trivial outcomes in this system that involves many components, we must start by minimizing the number of parameters, and identifying those complex processes, that control the evolution of the system. The free energy that we build on this biological system contains therefore out-of-equilibrium processes that can be approximated by a “close to equilibrium” description. Our approach is a classical one in statistical physics of active systems, namely in the effort to construct an equivalent free-energy for out-of-equilibrium systems. This allows to gain a clearer insight into those complex processes.

We have added a sentence in the main text, section ‘Momentum and free energy density derivation’, to clarify this point:

“Building a free-energy density for a biological material is justified, because, although biological materials are out of equilibrium, their behavior often resembles that dictated by thermodynamics. It is therefore useful to write a free energy in terms of state variables.”

Nevertheless, we recognize that we should have provided more tools for using our formalism by making it active. This is why we introduced the nematic order in the fibers in ‘Toward anisotropy’. This nematic order can be used to introduce active stress, and we have cited previous works by some of us see [Bell et al. 2024, Ackerman and Ben Amar 2023] as references for building active processes out of it.

We must also note that cell signaling has been introduced *a minima* in our system for providing the cue for the arrival of T-cells and NAFs from the boundaries. However, we found that although we had evoked the other role of the chemicals in the transformation from NAFs to CAFs in the text, details were not well explained. We have therefore corrected and added some explanations in the introduction of sections ‘Spatio-temporal behavior of tumor growth’, ‘Momentum and free energy density derivation’, and ‘Effective free energy density and source terms’.

A separate weakness concerns the assumption that fibroblasts affect T cell behavior primarily by just making a more dense ECM. There are a number of papers in the cancer literature (see, for some examples, Carstens, J., Correa de Sampaio, P., Yang, D. et al. Spatial computation of intratumoral T cells correlates with survival of patients with pancreatic cancer. Nat Commun 8, 15095 (2017);Sun, Xiujie, Bogang Wu, Huai-Chin Chiang, Hui Deng, Xiaowen Zhang, Wei Xiong, Junquan Liu et al. “Tumour DDR1 promotes collagen fibre alignment to instigate immune exclusion.” Nature 599, no. 7886 (2021): 673-678) that seem to indicate that density alone is not a sufficient indicator of T cell behavior. Instead, the organization of the ECM (for example, its anisotropy) could be playing a much more essential role than is given credit for here. This possibility is hinted at in the Discussion section but deserves much more emphasis.

The referee is right in his comment, and we thank him for raising this issue. We have therefore introduced the anisotropic orientation of the fibers, which induces an anisotropic friction in a new section ‘Toward anisotropy’. In addition, the references pointed out were included in this section. However, although the anisotropy strongly influences the fate of the tumor when the fibers are oriented perpendicular to the surface of the cancer nest, it is less effective when the fibroblasts are oriented in the direction of surface of the cancer nest. In the latter case, which is often the case before cancer cells reshape the tumor microenvironment, the matrix density should correlate with the friction.

Finally, the mixed version of the model is, from a general perspective, not very different from many other published models treating the ecology of the tumor microenvironment (for a survey, see Arabameri A, Asemani D, Hadjati J (2018), A structural methodology for modeling immune-tumor interactions including pro-and anti-tumor factors for clinical applications. Math Biosci 304:48-61). There are even papers in this literature that specifically investigate effects due to allowing cancer cells to instigate changes in other cells from being tumor-inhibiting to tumor-promoting. This feature occurs not only for fibroblasts but also for example for macrophages which can change their polarization from M1 to M2. There needed to be some more detailed comparison with this existing literature.

The referee is right that the first part of our approach, namely the dynamical system may be common in this kind of system, and it needs to be mentioned. So we added the following sentence in the discussion: “This is in line with several similar mathematical models, that study through this lens the inhibition/activation of the immune system by cancer cells either by means of compartmental nonlinear models similar to our dynamical system, for instance regarding macrophage recruitment and cytokine signaling (Arabameri et al., 2018; Li et al., 2019), or mixture models (Fotso et al., 2024). We combine the two approaches in order to rigorosly derive the parameters of the model and gain insights from both.”

**Recommendations for the authors:**

**Reviewer #1 (Recommendations for the authors):**
The authors should address the following points:Major issues(1) The shape of tumors simulated differs immensely from the observed tumors in Fig. 2. Here, the tumor is constituted by irregular domains, not dissimilar from domains in phase separating mixtures. The domains simulated are circular. Since the authors are using the space dependent model to model the increase in tumor cells with time in the different scenarios (immune-desert, immune-excluded, immune inflamed), it should explain how non-spherical tumor structures can be observed in these scenarios. The authors introduce tumor coalescence in page 28, however, it is not expected that the structures observed in Fig 2 are the result from different tumors merging and coalescing, because that would result from an unlikely large number of initial mutation events in the same region of the tissue. The authors should explain what mechanisms present in the model can lead to non-spherical forms.

We agree with the reviewer that real tumors are rarely round contrary to what our numerics suggests. In fact, only the last figure of our paper in the supporting information was more appropriate for such a discussion. We are now adding discussions and new figures to better illustrate our spatial model, see Figure 6 and ‘Toward anisotropy’. The in situ geometry of tumors depends on the shape of the host organ, the diffusive (chemical) or advected species such as T cells and fibroblasts, and on the nutrients. Thus, in our case, only cancer cells are produced locally, but during growth the tumor is strongly constrained by the microenvironment, and thus the geometry of the domain we model in the numerics and its boundary conditions. This is also true for the chemicals responsible for growth, cellular advection and phenotypic transformation. Their concentration depends on a convection-diffusion equation and boundary conditions. For a tumor in situ, such as in the lung, the available space is a constraint that will dominate the final geometry of the tumor nests. We do not think that coalescence is controlled by mutational events, but most likely by the search for space necessary for growth. Compared to the first version, we add new figures (Figure 6) that show that the geometry of the organ, as well as the localization of blood vessels, are a cause of the irregularity of the tumor shapes. We also introduce orientational order, which as suggested in section ‘Toward anisotropy’, can induce anisotropic friction and stresses, as well as anisotropic growth. We cite (Ackermann, Joseph, and Martine Ben Amar. “Onsager’s variational principle in proliferating biological tissues, in the presence of activity and anisotropy.” The European Physical Journal Plus 138.12 (2023): 1103.) where we described active stresses and coupling related to anisotropic growth.

(2) According to the authors, the model presented in equations (1) and onwards simulates the evolution of the fraction of tumor cells in the tissue. However the fraction of tumor cells, for example, depends itself on the variation of other cell types. For example, if fibroblasts were to proliferate with rate alpha, even without tumor cells proliferating, the fraction of tumor cells in the mixture should decrease as alpha times the tumor cells fraction. These terms are missing. The equations do not describe the evolution of the cells’ fractions but of the amount of cells of each type, normalised by the total carrying capacity of non-normal cells in the tissue. The text should be rewritten accordingly.

We agree with the referee: our definition of cell density was not precise enough and may appear misleading. In the paragraph II1, we more explictly introduce the word mass fraction which is the correct physical quantity to introduce into the spatial model.

“All these cells have the same mass density and the sum of their mass fraction satisfies the relationship *S = C + T + FNA + FA* = 1-*N*, where *N* is a healthy non active component as healthy cells, for example.”

It is less intuitive than “number of cells per unit volume” but necessary for the following (III)

(3) The authors start by calculating fixed points of different versions of the dynamical system without spatial dependence. They should explain what is the relevance of these fixed points: in a real situation, where the concentration of tumor fibroblasts and T-cells depend on position, in which conditions are these fixed points relevant?

The referee is right and we will clarify this point: the dynamic analysis is a help for understanding and predicting the scenario occurring in the system. After all the steps of paragraph 2.2, we are faced with 11 independent parameters only for the dynamical system and without the parameters generated by the space modeling itself. Our estimation concerns only lung cancer. These parameters do not appear in the literature. The parameters introduced in ‘Spatio-temporal behavior of tumor growth’ which are more related to physical interactions such as friction, cell-cell adhesion, etc. can be found in the literature or can be estimated and thus measured in in vitro experiments (see Ackermann and Ben Amar, EPJP 2023, P. Benaroch, J. Nikolic et al. 2024, biorxiv). So what are the fixed points for: they help to get the right numbers for spatial analysis. To recover special features of cancer evolution, we need a model, but also correct estimates of the data in a code that is quite technical and heavy, with each simulation taking a certain amount of time. For users who only need rough predictions, the analysis in ‘Dynamic modeling in the lung cancer TME’ is sufficient.

It is also important to note that the global result depends only on the source terms, and on the boundary conditions. This can be illustrated with a simple example: Consider the governing equation for the density of a component with velocity **v** and source term:

∂tϕ+∇⋅(ϕv)=γ

Integrating the equation over a fixed volume *V* of surface *S* gives:

∂tΦ=Γ−JS=Γ¯

Φ=∫ϕdV,Γ=∫γdΩ,JS=∫(ρv)⋅ndS. This integrated equation can then be approximated by the dynamical system that we write. Thus, while the dynamical system does not give any information about the local structure of the system, it may be indicative of its global outcome.

(4) In page 15, the authors identify that α_NA_ is proportional to δ𝝐^4^. However, in equation (7), they replace α_NA_ by δ𝝐^4^ without the proportionality constant. This should be corrected.

Thank you for your remark. This typo is now corrected.

(5) The tumor cell movement should be much slower than the T-cells. Here, the authors assign a similar friction coefficient for the cancer cells and T-cells, for example. However, in lung cancer tumor cells are epithelial, and adhere to each other in the tissue. Their movement is very restricted by the basement membranes and by cell-cell adhesion. Immune cells and T-cells on the other hand move rapidly throughout the stroma. It is a gross simplification to not consider the low epitelial tissue mobility in the context of lung cancer.

It is possible to assume different friction coe cients for each phase pair. This has been done in a previous publication, Ackermann et al., Physics report 2021. It is also possible to play with the cell-cell adhesion in the energy density and on the diffusion coe cient introduced in the Flory-Higgins free energy. Cell-cell adhesion is taken into account in the energy, and this makes the tumor a more dense phase, while T-cells can move towards cancer cells to which they are attracted. In the last part of the paper, we show the role of an anisotropic friction due to a nematic order for activated fibroblasts and all the other cells

(6) What is the biological mechanism by which the T-cells form a colony with a surface tension? In the phase-field model, the authors have a surface tension assigned to the cancer cells, T-cells and fibroblasts. Can the authors justify biologically why do they consider these surface tensions?

The fact that T-cells form a colony is due to the accumulation of T-cells at the outer boundary of the tumor, as they are attracted to it but cannot penetrate due to the strong cell-cell adhesion of the tumor cells in the nest. Adding a gradient square is standard in continuous models to limit the sharp variations. In a continuous approach, the gradient square contribution limits the sharp variations in cell density which are not physical.

Minor issues(a) Page 6 (end), characterisation of the fibre barrier produced by CAFs missing: what is the fibre density, how it can hinder the spread of cancer and T-cell motility? Is it so dense that it prevents ameboid movement? Can cells move through it using matrix degradation proteins?

The fiber density corresponds to the fibrous organic extracellular matrix secreted by cancer-associated fibroblasts. In desmotic (highly fibrous tumors such as PDAC or NSCLC), this extracellular matrix deposited around the tumor forms a physical barrier around the tumor nest, preventing both cell migration and capillary and immune cells penetration. In these cases, the fibrous belt actually prevents ameboid movement and cells must deform significantly to migrate. The role of this barrier was particularly demonstrated in the reference (Grout, John A., et al. “Spatial positioning and matrix programs of cancer-associated fibroblasts promote T-cell exclusion in human lung tumors.” Cancer Discovery 12.11 (2022): 2606-2625.). In later stages of cancer, the tumor may adapt and develop strategies to metastasize, such as matrix degradation. This matrix can be oriented, organized or disordered. To build a minimal model, we first considered an isotropic friction and also an anisotropic friction of the nematic belt, due to the activated fibroblasts. In the case of T-cells, as mentioned in section ‘Interactions between TME components', it is true that the biological literature also considers a phenotypic transformation of the T cells by the activated fibroblasts: this concerns both their proliferative capacities, antigen recognition and also their cytotoxic function. To better document the different mechanisms, we add the following publication: Cancer associated fibroblasts-an impediment to effective anti-cancer T cell immunity, by Koppensteiner, Lilian and Mathieson, Layla and O’Connor, Richard A and Akram, Ahsan R, Frontiers in immunology (2022).

However, our goal is to build a minimal model and to characterize and quantify the physical process in which CAFs are involved, namely the role of a physical barrier, that has been documented, as documented above.

(b) Page 19 (Fig 3), in the figure legend it is written “resting fibroblasts”, should be “non-activated fibroblasts”.

The referee is right: it will be better to write non-activated fibroblasts. This is now changed in the main text.

(c) Page 21 (equation), what is *d*Ω? It is *d***r**?

We thank the referee for raising this point. The text was indeed ambiguous as sometimes *d*Ω was replaced by *d***r**. To be clearer, all the elements of volume are now noted *dV* , and the element of surface of the system are noted *dS*.

In the article the units are in italic and should be in roman.

Thank you for raising this point. It has been corrected.

(d) Page 25 (beginning section ‘Ambiguous role of fibroblasts in tumorigenesis’), the authors mention that the simulation is 2D, however, the simulation has radial symmetry. A 1D simulation in radial coordinates could simulate a 3D spherical system. Is the simulation of this section equivalent to a 1D radial simulation (in 2D)?

The referee is right that in radial symmetry, a 1d equation may be written. We therefore present numerics with irregular shapes of the tumor nest in order to make the system fully 2d.

(e) Page 26 (Fig 4). Legends inside the plots of plates A, B, C and D are not clear. Colorbar range of plates A and D is different. Would facilitate if the ranges were the same.

The referee is right: the surface plots presented in figure 4 would be easier to compare with the same colorbar range for the legends. In fact, as the referee noted, figures in A, B and C have the same legends, while figure in D has a different one. This is due to the fact that D represents the case of the immune-inflamed tumor where the cancer mass fraction is quite vanishing, resulting in values that are of 3 orders of magnitude lower than those present in A, B and C. Therefore, they would disappear if the colorbar range were equal to the others.We insist more on the change of scale in the legend of Figure 4, in the new version.

(f) Page 29 (Fig 5), would facilitate if the order of immune-desert, immune-excluded, immune-inflamed was maintained throughout the document. In this figure the immune-inflamed case appears first.

We agree with the reviewer that following the same order in which the different cases are presented throughout the manuscript would be helpful in comparing the different figures. Therefore, we have modified Figure 5.

(g) Page 31, the authors indicate that pharmacodynamics and pharmacokinetics are highly dependent on tumour spatial structure. Can they provide examples and citations?

In the discussion, we have added references concerning pharmacodynamics.

(h) Page 33 (Fig Sup2), would facilitate if the order of immune-desert, immune-excluded, immune-inflamed was maintained throughout the document.

We thank the reviewer for pointing this out, the order of the different scenarios in Fig Sup 2 has now been changed.

**Reviewer #2 (Recommendations for the authors):**
Major points(1) Following on from the discussion in the public review, I feel that there are a number of critical issues that need to be addressed regarding modeling assumptions. I would like to understand why the authors believe it is possible to use a free energy-driven model of the microenvironment when many of the processes relevant for their study have an undeniably “active media” flavor.

The referee is right that processes in biology are active processes. However, it is a classical approach to model physical interactions between biological components with a free-energy, especially cell adhesion, as they often lead to quasi-stationary equilibrium-like patterns. The free-energy approach has also the advantage to derive straight-forwardly complex phenomena involving many components. Activity can indeed be introduced in such a framework, if we know that the fibroblasts transform into myo-fibroblasts, see for example our previous publication Ackermann and Ben Amar, EPJP 2023. However, in the interest of simplification and reduction of the number of free parameters, we have not not considered further complication of the model here, as a minimal model allows to distinguish the main processes that occur. Nevertheless, introducing more precisely activity, in the nematic approach already achieved for the friction, is a natural continuation of our work: See the new section ‘Toward anisotropy’, where we introduce the nematic order, and we indicate that active nematic stresses can be written from it.

Next, I don’t understand the assumption that T cells do not proliferate once they detect neoantigens on the cancer cells; activation of T cells usually causes them to become more proliferative.

We thank the referee for this question. The T-cell fraction has two origins: proliferation of T-cells in situ in the stroma or inside tumor nest or external arrival from the sources that we privilege. We recognize that a full analysis of the tumor-microenvironment would require to consider proliferation near the tumor, as many more other processes which is do able but requires the knowledge of more biological date. In addition, besides, the proliferation of T-cells will be equivalent to increase the killing abilities of T-cells and these two effect overlapp in our approach.

In order to clarify this point, we modify the following sentence in ‘Dynamical system for immune and cancer cells in interaction’:

“Although proliferation of cytotoxic T-cells has been observed, we do not consider explicitly proliferation in our study as we focus on their ability to infiltrate the tumor.”

Rather, we consider that T-cells proliferate outside the domain boundaries, so that this proliferation is included in the boundary source contributions.

Finally, the issue of whether the density of fibers is sufficient to understand the role of fibroblasts is not at all settled. There should be a full discussion of this issue including mentioning of the Nature paper (cited in the public review) that argues that orientation (and not density) is the key to the role of fibers, as well as the earlier cited work of Kalluri and collaborators on the role of ECM density in pancreatic cancer.

We thank the referee for this remark. As we wrote above in the response to the public review, we introduced significant additions that aim to tackle this question in the article.

(2) The authors present a picture of a tumor cell with fibroblasts apparently arrayed circumferentially around the tumor boundary and therefore blocking infiltration. This type of tumor structure has been seen before, for example in “On the mechanism of long-range orientational order of fibroblasts.” Proceedings of the National Academy of Sciences 114, no. 34 (2017): 8974-8979, which should be cited. More importantly, in that paper the argument is made that positive feedback between fibroblasts and ECM geometry can cause structures like this to form. If this is indeed what is occurring, this would indicate the crucial importance of a mechanism beyond what is contained in the current model. This issue should therefore be discussed within this paper. This issue is of course connected to the previous point regarding the role of ECM structure beyond density.

We completely agree that the interplay between the fibroblast layer and the tumor shapes the tumor boundary. One of the authors has worked recently on this precise topic (Aging and freezing of active nematic dynamics of cancer-associated fibroblasts by fibronectin matrix remodeling, C Jacques, J Ackermann, S Bell, C Hallopeau, CP Gonzalez, ... bioRxiv, 2023.11. 22.568216, Ordering, spontaneous flows and aging in active fluids depositing tracks S Bell, J Ackermann, A Maitra, R Voituriez arXiv preprint arXiv:2409.05195). Since the fibroblast layer is an active material, it contributes to an anisotropic stress that can be introduced into the model. Our first strategy was to present the simplest modeling in order to focus on the most important interactions as cell-cell adhesion and cell-tissue adhesion. However, we recognize that those questions should be discussed in the text, and we discuss it in the new section ‘Toward anisotropy’.

Minor pointsThere are also a number of more minor points to consider:(1) Since the parameter is taken to be O(1), why exactly does it matter how the other parameters scale with it?

It is very important to compare the order of magnitude of the other parameters once the selected parameter of order O(1) is really the driving parameter of the coupling. It gives a first picture of the main interactions that has to consider.

(2) I didn’t understand the relevance of referring specifically to IL 6 among many other possibly relevant signals, as is currently done on page 7.

This corresponds to studies aiming to correlate lung cancer risks and the concentration of interleukin, mostly IL6 and IL8 (McKeown, D. J., et al. ”The relationship between circulating concentrations of C-reactive protein, inflammatory cytokines and cytokine receptors in patients with non-small-cell lung cancer.” British journal of cancer 91.12 (2004): 1993-1995.,Brenner, Darren R., et al. ”Inflammatory cytokines and lung cancer risk in 3 prospective studies.” American journal of epidemiology 185.2 (2017): 86-95.) but in the absence of very detailed biological information, the modeling and its results are not modified if other chemicals intervene..We slightly modified the following phrase in section ‘Interactions between TME components’:

“In particular, in the family of inflammatory proteins, also called cytokines, Interlukin-6 (IL6) and (IL8) seem, among others to stimulate the infiltration of CD8^+^.

(3) The authors need to mention the possibility of T-cell chemotaxis to the tumor being “self-amplified” in the T cell system, as put forth in Galeano Ninõ, Jorge Luis, Sophie V. Pageon, Szun S. Tay, Feyza Colakoglu, Daryan Kempe, Jack Hywood, Jessica K. Mazalo et al. “Cytotoxic T cells swarm by homotypic chemokine signalling.” eLife 9 (2020): e56554. This might again reveal a needed extension of the current modelling strategy.

We thank the referee for his/her comment on the self-amplification of T-cell population in the stroma and we mention the indicated reference in our paper. This auto-chemoatactic process which induces a dynamic of more e cient recruitment towards the tumor, may be important for immunotherapy. To have more e cient T-cell arriving at the site of the tumor, will lead a better issue for the patient, if the swarming organization is maintained in a desmoplastic nematic stroma.

(4) It is not obvious to me that in sub figures 3F and 3H the tumor is enroute to being totally eradicated, as is stated in the text. The blue lines seemed to asymptote at non-zero population values.

Looking at sub-figures 3F and 3H, we stated in the main text that the tumor is eradicated as the representative population approaches a 0 value fraction, or at least decays around the 0 (0.01/0.05 to be more precise). This is even more evident when compared with the other cases where the tumor mass fraction reaches values of a higher order (up to 0.6), thus leading us to dinstinguish between these different scenarios.

(5) The description of the interaction of cells with fibers as being increased friction might be misleading, as the real effect could be actual trapping in the network (as opposed to just slowing down the motion).

We thank the referee for this question as it allow us to make an important distinction. Indeed, what the referee describes seems to correspond to a discrete event, namely a cell trapped in a network. However, coarse-graining the dynamics to the continuous modeling seems to us as leading to an effective friction between the two phases. Moreover, we also now introduced an anisotropic friction which can represent a trapping. The velocities are not only directed around the tumor but can also be oriented towards the tumor, so that eventually the friction along the radius mimics a trapping (see Fig.4 on top). We have introduced this anisotropic friction via a nematic model, see the appendix.